# Cryo-EM structures reveal high-resolution mechanism of a DNA polymerase sliding clamp loader

**Christl Gaubitz[1], Xingchen Liu[1], Joshua Pajak[1], Nicholas P Stone[1], Janelle A Hayes[1], Gabriel Demo[2], Brian A Kelch[1]\***

[1]Department of Biochemistry and Molecular Biotechnology, University of Massachusetts Chan Medical School, Worcester, United States; [2]RNA Therapeutics Institute, University of Massachusetts Chan Medical School, Worcester MA & Central European Institute of Technology, Masaryk University, Brno, Czech Republic

**Abstract** Sliding clamps are ring-shaped protein complexes that are integral to the DNA replication machinery of all life. Sliding clamps are opened and installed onto DNA by clamp loader AAA+ ATPase complexes. However, how a clamp loader opens and closes the sliding clamp around DNA is still unknown. Here, we describe structures of the *Saccharomyces cerevisiae* clamp loader Replication Factor C (RFC) bound to its cognate sliding clamp Proliferating Cell Nuclear Antigen (PCNA) en route to successful loading. RFC first binds to PCNA in a dynamic, closed conformation that blocks both ATPase activity and DNA binding. RFC then opens the PCNA ring through a large-scale 'crab-claw' expansion of both RFC and PCNA that explains how RFC prefers initial binding of PCNA over DNA. Next, the open RFC:PCNA complex binds DNA and interrogates the primer-template junction using a surprising base-flipping mechanism. Our structures indicate that initial PCNA opening and subsequent closure around DNA do not require ATP hydrolysis, but are driven by binding energy. ATP hydrolysis, which is necessary for RFC release, is triggered by interactions with both PCNA and DNA, explaining RFC's switch-like ATPase activity. Our work reveals how a AAA+ machine undergoes dramatic conformational changes for achieving binding preference and substrate remodeling.

**\*For correspondence:**
brian.kelch@umassmed.edu

**Competing interest:** The authors declare that no competing interests exist.

## Editor's evaluation

Clamp loader-sliding clamp complexes are required for DNA replication and repair in all domains of life. This study reports several cryo-EM structures of multiple clamp loading intermediates from a single species, *Saccharomyces cerevisiae*, providing new insights into the mechanism of clamp loading and the ligand-induced conformational dynamics of molecular motors and switches.

## Introduction

In all known cellular life, DNA replication is coordinated by ring-shaped sliding clamp proteins that wrap around DNA to activate DNA polymerases and other factors (*Moldovan et al., 2007*). Sliding clamps are regulated by their presence on DNA, which in turn is governed by clamp loaders that open the sliding clamp ring and place it onto DNA (*Kelch, 2016*). The clamp loader of eukaryotes Replication Factor C (RFC) installs the sliding clamp Proliferating Cell Nuclear Antigen (PCNA) in a coordinated and stepwise fashion (*Kelch, 2016*). First, RFC binds ATP, which is a prerequisite for tight binding to PCNA (*Sakato et al., 2012a*). Next, RFC binds to PCNA, and then opens the PCNA ring. This open ternary complex is now competent to bind to primer–template (p/t) DNA (double-stranded DNA with a single-stranded 5' overhang). Primer–template binding to the ternary complex triggers

**Figure 1.** Architecture of the eukaryotic clamp loader (RFC) and clamp (PCNA). (**A**) RFC is composed of five different subunits (named A–E) that each consist of the AAA+ ATPase module and a collar domain. The nucleotide-binding site is sandwiched between the N-terminal Rossmann fold domain and the Lid domain of the ATPase module at the subunit interface. The ATPase module and a C-terminal extension of the A subunit called the A′-domain form the A-gate. (**B**) Domain organization of RFC subunits. (**C**) Clamp loading begins with binding of ATP to RFC, followed by PCNA binding. How PCNA is opened and DNA binds to the open RFC:PCNA complex is not known. DNA-binding triggers ATP hydrolysis, PCNA closure, and RFC ejection. Structures obtained prior for RFC:PCNA complexes are indicated (*Bowman et al., 2004*; *Gaubitz et al., 2020*).

The online version of this article includes the following source data and figure supplement(s) for figure 1:

**Figure supplement 1.** Characterization and cryo-EM of full-length RFC:PCNA.

**Figure supplement 1—source data 1.** Image of SDS-PAGE gel of fractions from gel filtration of RFC and PCNA.

**Figure supplement 2.** Schematic of yRFC:PCNA cryo-EM processing.

**Figure supplement 3.** Schematic of yRFC:PCNA:DNA cryo-EM processing.

ATP hydrolysis in the clamp loader, followed by sliding clamp closure and ultimately release of the clamp loader complex (*Chen et al., 2009*). Therefore, RFC has two macromolecular substrates, PCNA and p/t-DNA, that must bind sequentially. Yet how clamp loaders achieve this strict sequential order remains unknown.

Clamp loaders are members of the <u>A</u>TPases <u>a</u>ssociated with various cellular <u>a</u>ctivities (AAA+) family (*Erzberger and Berger, 2006*). Most members of this family function as ringed, homohexameric molecular motors that harvest energy from ATP to translocate substrates through their central pore (*Jessop et al., 2021*). However, clamp loaders are not motors but instead are ATP-dependent remodeling switches (*Kelch, 2016*). In contrast to typical AAA+ ATPases, clamp loaders are heteropentameric with the five different subunits called A–E (*Figure 1A, B*). Each subunit features a classic AAA+ ATPase module, which holds the ATP sandwiched between the Rossmann fold and Lid domain at the binding interface with the neighboring subunit. The AAA+ modules of every subunit are extended by collar domains, which tightly associate together into a flat disk, enabling dynamic interactions between the five AAA+ modules.

In addition to the canonical AAA+ machinery, many clamp loaders contain an A′ domain that bridges the gap between the A and E subunits. The space between the A′ domain and the AAA+ domain of

**Table 1.** Clamp loader structures previously obtained for the various states in the clamp loading cycle.

**Clamp loader prior to clamp binding**

| Species | Composition | Reference | PDB accession number |
|---|---|---|---|
| *E. coli* | Clamp loader alone | *Jeruzalmi et al., 2001a* | 1JR3 |
| *E. coli* | Clamp loader, ADP | *Kazmirski et al., 2004* | 1XXI |
| *E. coli* | Clamp loader, ATP analog | *Kazmirski et al., 2004* | 1XXH |
| *E. coli* | Clamp loader, ATP analog, primer/template DNA | *Simonetta et al., 2009* | 3GLF |
| **Encounter complex of clamp loader bound to the closed clamp** | | | |
| *H. sapiens* | Clamp loader bound to the clamp, ATP analog | *Gaubitz et al., 2020* | 6VVO |
| *S. cerevisiae* | Clamp loader bound to the closed clamp, ATP analog | *Bowman et al., 2004* | 1SXJ |
| **Clamp loader bound to the clamp and primer/template DNA** | | | |
| T4 phage | Clamp loader, open clamp, ATP analog, DNA | *Kelch et al., 2011* | 3U60 |
| T4 phage | Clamp loader, closed clamp, ATP analog, DNA | *Kelch et al., 2011* | 3U5Z |
| T4 phage | Clamp loader, closed clamp, ATP analog, ADP, DNA | *Kelch et al., 2011* | 3U61 |

subunit A is the 'A-gate' (*Figure 1C*), which serves as the entry site for p/t-DNA binding. It was initially proposed that ATP-binding triggers the five AAA+ modules to form a spiral with a symmetrical pitch that matches the geometry of DNA and templates the open clamp (*Bowman et al., 2004*; *Simonetta et al., 2009*, *Table 1*). This symmetric, helical arrangement of the subunits results in a cracked interface between the A and E subunits, bridged by the A' domain. As the A' domain stretches away from the A subunit to maintain contact, the A-gate opens and permits p/t-DNA binding (*Kelch et al., 2011*). However, structures of the human and yeast RFC:PCNA complexes bound to ATP analog show a closed PCNA ring bound to RFC in an autoinhibited state, where the closed A-gate blocks the DNA binding (*Bowman et al., 2004*; *Gaubitz et al., 2020*, *Table 1* ). Additionally, another element called the 'E-plug' reaches into RFC's central chamber and sterically occludes DNA binding. This autoinhibited state of RFC bound to closed PCNA is likely the first intermediate in the clamp loading reaction (*Gaubitz et al., 2020*; *Sakato et al., 2012a*; *Thompson et al., 2012*).

The question remains: How does the clamp loader open the sliding clamp? This is perhaps the most important function of the clamp loader, yet clues as to how this process is achieved remain elusive (*Figure 1C*). The structure of the T4 phage loader bound to DNA and an open clamp indicated that the clamp adopts a right-handed spiral conformation that matches the helical pitch of DNA (*Kelch et al., 2011*). However, this structure represents the state after DNA is bound (*Table 1*), and does not address how the clamp ring is initially opened. Thus, the structure of a clamp loader bound to an open clamp without DNA has been sought after, as it will illuminate the opening process.

## Results

### Structures of RFC:PCNA complexes en route to DNA loading

To understand how RFC opens PCNA and subsequently binds DNA, we used single-particle cryo-EM to determine structures of full-length *Saccharomyces cerevisiae* RFC bound to PCNA and the slowly hydrolyzing ATP analog ATPγS in the presence and absence of primer–template (p/t) DNA. We reconstituted the complex from purified RFC and PCNA subcomplexes that were separately expressed in *E. coli* (*Figure 1—figure supplement 1A*). Full-length RFC is functional, as it has the expected ATPase activity profile (*McNally et al., 2010*) with PCNA and p/t-DNA synergistically activating ATP hydrolysis (Figure 6F).

To prevent particle denaturation during sample preparation for cryo-EM, we crosslinked DNA-free and DNA-bound complexes using the amine-reactive crosslinker bis(sulfosuccinimidyl)suberate (BS3). Mild crosslinking is frequently used to obtain high-resolution cryo-EM structures of labile complexes (*Gerlach et al., 2018*; *Yoo et al., 2018*; *Gaubitz et al., 2020*). Mass spectrometry of the DNA-free sample reveals that most crosslinks are intramolecular and map to the unresolved N- and C-termini of

RFC1, with only a few detectable intermolecular crosslinks between RFC subunits (*Figure 1—figure supplement 1B*; *Table 2*). No significant crosslinks were observed between RFC and PCNA.

We imaged the RFC:PCNA complex with and without p/t-DNA using a 300 kV Titan Krios microscope (*Figure 1—figure supplement 1C, D*, *Figure 2A. B*, and *Figure 3A, B*; *Table 3*). 3D classification results in four well-defined reconstructions from the DNA-free sample, with overall resolutions ranging between 3.8 and 4.0 Å (*Figure 1—figure supplement 2D*). The dataset of the DNA-containing sample yielded several well-defined classes, with overall resolutions ranging between 3.3 and 3.5 Å (*Figure 1—figure supplement 3C*, *Table 3*). We focused on classes in which all subunits of RFC and PCNA are visible, although the N- and C-terminal regions of the A subunit lack clear density. The quality of the cryo-EM reconstructions readily permitted model building using the crystal structure as a template (*Bowman et al., 2004*; *Figure 1—figure supplement 1C, D*; *Table 3*).

## The initial complex of RFC:PCNA is dynamic

Three of the classes from the DNA-free sample are of RFC bound to closed PCNA in different conformational states (*Figure 2*). Overall, these structures resemble the previous yeast RFC:PCNA crystal structure and our recent cryo-EM structure of human RFC (hRFC):PCNA (*Figure 2—figure supplement 1A*; *Bowman et al., 2004*; *Gaubitz et al., 2020*). The PCNA ring is closed with only the A, B, and C subunits of RFC contacting PCNA (*Figure 2B*). The interaction area between clamp loader and clamp averages ~1940 Å$^2$ across the three states. The nucleotide density in each of the four active sites is most consistent with the presence of ATPγS, although the density for the γ-phosphate analog in the D subunit is somewhat ambiguous due to low local resolution throughout this subunit. Nonetheless, the ATPase sites of the B, C, and D subunits are in an inactive state (*Figure 2—figure supplement 1B*), with the AAA+ spiral in the overtwisted state observed in the hRFC structure and the previous yeast RFC crystal structure (*Figure 2—figure supplement 1C*; *Bowman et al., 2004*; *Gaubitz et al., 2020*). Therefore, all three of these structures represent autoinhibited states of RFC (termed Autoinhibited1, Autoinhibited2, and Autoinhibited3). Because the Autoinhibited1, 2, and 3 states likely represent ATP-saturated configurations, we place these conformational states early in the clamp loading reaction.

The subunits in the AAA + spiral have a different tilt in each of the Autoinhibited states, thereby slightly altering the intersubunit interactions (*Figure 2—figure supplement 1C*). For instance, the Autoinhibited3 state exhibits a slightly cracked A-gate (but not open enough for DNA to pass through), whereas the A-gate is closed in the Autoinhibited1 and 2 states (*Figure 2A*, *Figure 2—figure supplement 2A*). Further, the AAA + modules of subunits C and D change their position into a more symmetric alignment with overlapping rotation axes relative to Autoinhibited1 and 2 (*Figure 2—figure supplement 1C*). On the other hand, the PCNA ring tilts ~19° relative to the RFC-D in the Autoinhibited2 state relative to the Autoinhibited1 and 3 states (*Figure 2A*, *Figure 2—figure supplement 2A*).

Despite these differences, the three Autoinhibited structures are very similar, and so we asked if these conformations represent distinct intermediates or if they are snapshots along a continuum of conformations. Therefore, we characterized the particles that contribute to the Autoinhibited states using multibody refinement (*Figure 2—figure supplement 2B–I*), a computational tool that allows modeling of macromolecular motion (*Nakane et al., 2018*). To examine motion between clamp and clamp loader, we defined RFC and PCNA as two independent rigid bodies (*Figure 2—figure supplement 2B–E*). This analysis revealed that the dominant motion is rocking of PCNA toward RFC, with the linker between the ATPase and collar domains serving as a hinge (*Figure 2C*, *Video 1*). Other motions include swiveling of the RFC spiral with RFC-D getting closer to PCNA (*Figure 2—figure supplement 2E*, *Video 2*). These results are not dependent on the particular mask used, as similar motions are observed using different masking strategies (*Figure 2—figure supplement 2F–I*). Principal component analysis of the multibody conformers revealed a unimodal distribution of particles along their eigenvalue (*Figure 2D*, *Figure 2—figure supplement 2D,H*). This unimodal distribution indicates that the three different observed cryo-EM class averages do not represent particles in discrete states, but rather snapshots along a continuum of motion. Thus, the autoinhibited state of RFC is conformationally heterogeneous, with the dominant motions driving RFC toward PCNA. We propose these motions represent an early phase of the transition toward opening of the PCNA ring.

**Table 2.** List of BS3 crosslinks.

| XlinkX score | Type | # Crosslink spectral matches | Sequence A | Position A | Sequence B | Position B | Protein A | Protein B |
|---|---|---|---|---|---|---|---|---|
| 58,66 | Inter | 1 | [K]LHLPPGK | 100 | [K]LAATR | 274 | RFC4 | RFC1 |
| 58,64 | Inter | 3 | [K]LELNVVSSPYHLEITPSDMGNNDR | 82 | S[K]TLLNAGVK | 385 | RFC5 | RFC1 |
| 56,99 | Inter | 3 | [K]YVNTFMK | 285 | DIL[K]R | 220 | RFC2 | RFC5 |
| 56,47 | Inter | 1 | NQI[K]DFASTR | 98 | [K]LAATR | 274 | RFC3 | RFC1 |
| 52,59 | Inter | 2 | E[K]VKNFAR | 109 | TME[K]YSK | 160 | RFC2 | RFC5 |
| 50,97 | Inter | 1 | NQI[K]DFASTR | 98 | RPDANSI[K]SR | 484 | RFC3 | RFC1 |
| 48,17 | Inter | 1 | GASEALA[K]R | 182 | [K]IVKER | 269 | RFC1 | RFC5 |
| 45,16 | Inter | 1 | YT[K]NTR | 139 | [K]EEER | 267 | RFC3 | RFC1 |
| 41,65 | Inter | 1 | [K]LEEQHNIATK | 249 | YT[K]NTR | 139 | RFC1 | RFC3 |
| 91,6 | Intra | 3 | [K]LEEQHNIATK | 249 | RPDANSI[K]SR | 484 | RFC1 | RFC1 |
| 72,73 | Intra | 4 | EAELLV[K]KEEER | 266 | [K]LAATR | 274 | RFC1 | RFC1 |
| 71,87 | Intra | 12 | QLIAGMPAEGGDGEAAE[K]AR | 245 | R[K]LEEQHNIATK | 249 | RFC1 | RFC1 |
| 71,27 | Intra | 2 | E[K]FKLDPNVIDR | 495 | [K]LAATR | 274 | RFC1 | RFC1 |
| 71,03 | Intra | 1 | F[K]LDPNVIDR | 497 | [K]LAATR | 274 | RFC1 | RFC1 |
| 71,03 | Intra | 9 | [K]TSTPLILICNER | 446 | S[K]TLLNAGVK | 385 | RFC1 | RFC1 |
| 64 | Intra | 1 | EAELLV[K]KEEER | 266 | S[K]KLAATR | 273 | RFC1 | RFC1 |
| 62,71 | Intra | 1 | RPDANSI[K]SR | 484 | SA[K]YYR | 678 | RFC1 | RFC1 |
| 62,2 | Intra | 2 | YAPTNLQQVCGN[K]GSVMK | 314 | L[K]NWLANWENSKK | 321 | RFC1 | RFC1 |
| 61,3 | Intra | 4 | EAELLVK[K]EEERSK | 267 | [K]LAATR | 274 | RFC1 | RFC1 |
| 60,15 | Intra | 1 | FAFACNQSN[K]IIEPLQSR | 149 | VT[K]NLAQVK | 275 | RFC4 | RFC4 |
| 60,15 | Intra | 3 | YS[K]LSDEDVLKR | 165 | VT[K]NLAQVK | 275 | RFC4 | RFC4 |
| 58,98 | Intra | 1 | IPATV[K]SGFTR | 767 | HAG[K]DGSGVFR | 340 | RFC1 | RFC1 |
| 58,55 | Intra | 4 | GASEALA[K]R | 182 | VT[K]SISSK | 190 | RFC1 | RFC1 |
| 57,1 | Intra | 3 | RPDANSI[K]SR | 484 | [K]EEER | 267 | RFC1 | RFC1 |
| 56,99 | Intra | 1 | KLEEQHNIAT[K]EAELLVK | 259 | [K]EEER | 267 | RFC1 | RFC1 |
| 56,99 | Intra | 1 | DNVVREED[K]LWTVK | 296 | [K]EEER | 267 | RFC1 | RFC1 |
| 56,41 | Intra | 1 | [K]YNSMTHPVAIYR | 773 | LGTSTD[K]IGLR | 698 | RFC1 | RFC1 |
| 53,33 | Intra | 1 | Y[K]CVIINEANSLTK | 136 | L[K]IDVR | 69 | RFC5 | RFC5 |
| 52,59 | Intra | 2 | [K]ASSPTVKPASSK | 77 | [K]TKPSSK | 90 | RFC1 | RFC1 |
| 52,59 | Intra | 2 | HAG[K]DGSGVFR | 340 | GSVM[K]LK | 319 | RFC1 | RFC1 |
| 52,59 | Intra | 2 | ASSPTV[K]PASSK | 84 | [K]TKPSSK | 90 | RFC1 | RFC1 |
| 51,79 | Intra | 2 | [K]LEEQHNIATK | 249 | [K]LAATR | 274 | RFC1 | RFC1 |
| 50,97 | Intra | 1 | [K]TATSKPGGSK | 845 | S[K]TLLNAGVK | 385 | RFC1 | RFC1 |
| 50,34 | Intra | 1 | KMPVSNVIDVSETPEGE[K]K | 68 | LPLPA[K]R | 75 | RFC1 | RFC1 |
| 49,59 | Intra | 4 | EKF[K]LDPNVIDR | 497 | RPDANSI[K]SR | 484 | RFC1 | RFC1 |
| 47,92 | Intra | 1 | LGTSTD[K]IGLR | 698 | [K]LAATR | 274 | RFC1 | RFC1 |
| 47,92 | Intra | 1 | S[K]TLLNAGVK | 385 | [K]LAATR | 274 | RFC1 | RFC1 |
| 47,92 | Intra | 1 | GASEALA[K]R | 182 | [K]LAATR | 274 | RFC1 | RFC1 |
| 47,85 | Intra | 2 | SISS[K]TSVVLGDEAGPK | 195 | [K]LEEQHNIATK | 249 | RFC1 | RFC1 |
| 47,85 | Intra | 1 | [K]YNSMTHPVAIYR | 773 | [K]TATSKPGGSK | 845 | RFC1 | RFC1 |

*Table 2 continued on next page*

*Table 2 continued*

| XlinkX score | Type | # Crosslink spectral matches | Sequence A | Position A | Sequence B | Position B | Protein A | Protein B |
|---|---|---|---|---|---|---|---|---|
| 46,57 | Intra | 4 | R[K]LEEQHNIATK | 249 | GASEALA[K]R | 182 | RFC1 | RFC1 |
| 46,35 | Intra | 1 | [K]ASSPTVKPASSK | 77 | VT[K]SISSK | 190 | RFC1 | RFC1 |
| 45,16 | Intra | 1 | YAPTNLQQVCGN[K]GSVMK | 314 | [K]EEER | 267 | RFC1 | RFC1 |
| 45,16 | Intra | 1 | E[K]FKLDPNVIDR | 495 | [K]EEER | 267 | RFC1 | RFC1 |
| 45,16 | Intra | 2 | [K]LEEQHNIATK | 249 | [K]EEER | 267 | RFC1 | RFC1 |
| 44,72 | Intra | 1 | E[K]FKLDPNVIDR | 495 | RPDANSI[K]SR | 484 | RFC1 | RFC1 |
| 44,45 | Intra | 1 | YAPTNLQQVCGN[K]GSVMK | 314 | [K]LEEQHNIATK | 249 | RFC1 | RFC1 |
| 44,14 | Intra | 2 | NLP[K]MRPFDR | 462 | S[K]TLLNAGVK | 385 | RFC1 | RFC1 |
| 44,14 | Intra | 1 | RPDANSI[K]SR | 484 | GASEALA[K]R | 182 | RFC1 | RFC1 |
| 44,12 | Intra | 1 | [K]LEEQHNIATK | 249 | [K]TKPSSK | 90 | RFC1 | RFC1 |
| 43,7 | Intra | 1 | NLP[K]MRPFDR | 462 | LGTSTD[K]IGLR | 698 | RFC1 | RFC1 |
| 43,7 | Intra | 1 | [K]YNSMTHPVAIYR | 773 | TATS[K]PGGSK | 850 | RFC1 | RFC1 |
| 41,98 | Intra | 2 | LGTSTD[K]IGLR | 698 | RPDANSI[K]SR | 484 | RFC1 | RFC1 |
| 41,98 | Intra | 1 | [K]LEEQHNIATK | 249 | F[K]LDPNVIDR | 497 | RFC1 | RFC1 |
| 41,98 | Intra | 1 | HAG[K]DGSGVFR | 340 | VT[K]SISSK | 190 | RFC1 | RFC1 |
| 41,94 | Intra | 1 | NQI[K]DFASTR | 98 | YT[K]NTR | 139 | RFC3 | RFC3 |
| 40,95 | Intra | 1 | NLAQV[K]ESVR | 281 | IHKLNN[K]A | 322 | RFC4 | RFC4 |
| 40,92 | Intra | 1 | KLPLPA[K]R | 75 | [K]EEER | 267 | RFC1 | RFC1 |

## PCNA opening is coupled to large-scale expansion of RFC

Each of the two cryo-EM datasets revealed a class of RFC bound to open PCNA with no DNA bound (*Figure 1—figure supplements 2D and 3C*). To our knowledge, these are the first high-resolution structures of a clamp loader bound to an open clamp prior to DNA binding. Both reconstructions are highly similar (overall $C_\alpha$ RMSD is 0.74 Å, map to map correlation coefficient is ~0.85) and we refer to these structures as Open1 and Open2 (*Figure 3—figure supplement 1A–D*). PCNA forms a right-handed spiral with a ~20 Å opening that is wide enough for dsDNA to enter (*Figure 3A, B*). The PCNA ring opens primarily through in-plane rather than out-of-plane motions (in-plane ~19 Å and out-of-plane ~10 Å for Open2, *Figure 3C*). Each of the subunits of PCNA twists outward and toward RFC, with the largest distortion in subunit II (*Figure 3—figure supplement 1F*).

PCNA opens at the A-gate of RFC, disrupting the interaction between the first and third subunits of the PCNA ring (termed PCNA-I and PCNA-III, hereafter). The open PCNA ring is directly held by all five subunits of RFC, burying ~3800 Å² of surface area, an approximate ~1860 Å² increase over that of the Autoinhibited states (*Figure 3B*, *Figure 3—figure supplement 2A, B*). The RFC-C subunit shifts downward to interact much more tightly with PCNA-II, while PCNA also forms new interactions with RFC-D, RFC-E, and the A' domain of RFC-A (*Figure 3—figure supplement 2C, D*). The overall interface is characterized by an alternating pattern of strong and weak interactions (strong: RFC-A, -C, and -E; weak: RFC-B, -D, and A'). The strong interactions are with the main partner binding pocket of PCNA, using a binding region that resembles a common motif for PCNA-interacting partners. Of these strong interfaces, RFC-A is the most substantial and RFC-E weakest; RFC-A contains a true PCNA interaction motif, while RFC-C and RFC-E's motifs are increasingly degenerate. It is likely that the stronger interactions at the 'bottom' of the spiral allows the clamp loader to toggle between the closed and open states of PCNA without releasing RFC.

The AAA+ modules of RFC adopt a right-handed spiral whose periodicity matches that of the six contact sites on PCNA. The symmetry of the ATPase spiral can be visualized by the near perfect alignment of the rotation axes that relate adjacent AAA+ subunits (*Figure 3F, G*). The interfaces between adjacent AAA+ modules become tighter, bringing the catalytic arginine finger residue closer

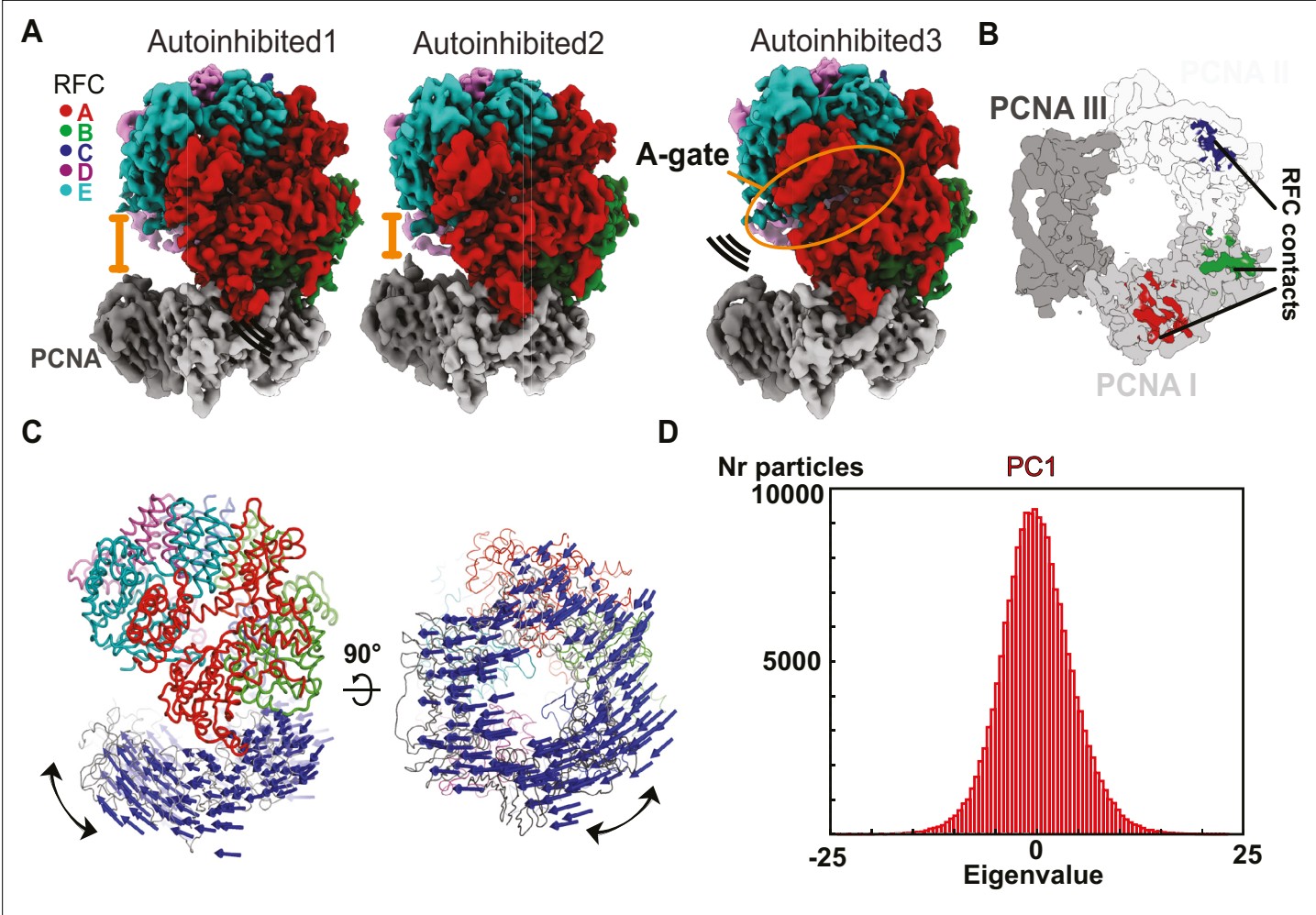

**Figure 2.** The Autoinhibited state is dynamic. (**A**) Cryo-EM maps of the three Autoinhibited conformations of the RFC:PCNA complex. PCNA tilts closer relative to RFC in Autoinhibited2. The subunit arrangement of the AAA+ module of Autoinhibited3 is changed slightly, which leads to a crack in the A-gate. (**B**) Top view on the contact sites of PCNA with RFC in the autoinhibited conformation. (**C**) Principal component analysis of all Autoinhibited particles reveals a rocking motion of PCNA relative to RFC. The $C_\alpha$ displacement of principal component 1 (PC1) is indicated by arrows, scaled down by a factor of 2. (**D**) Principal component analysis reveals a range of motions within the initial RFC:PCNA complex. Amplitude histogram of the first principal component (PC1) reveals a unimodal distribution of particles, suggesting that this state consists of related particles in continuous motion.

The online version of this article includes the following figure supplement(s) for figure 2:

**Figure supplement 1.** RFC:PCNA complexes in autoinhibited conformations.

**Figure supplement 2.** Multibody analyses with all 183,571 particles combined from the three Autoinhibited states to investigate the dynamic initial complex of RFC with PCNA.

to the neighboring ATPase site and potentiating ATP hydrolysis. This observation explains the modest boost in ATP activity upon PCNA binding (*Johnson et al., 2006*; Figure 6F). However, similar to the Autoinhibited structures, all four active sites remain bound to ATP analog (*Figure 3—figure supplement 1*). Therefore, while opening is necessary to promote ATP hydrolysis by properly positioning the *trans*-acting arginine finger residues across the intersubunit interface, ATP hydrolysis is not necessary to drive the conformational change from Autoinhibited to Open and opening is likely not sufficient to stimulate ATP hydrolysis on its own.

In order to rupture the PCNA ring, the AAA+ spiral of RFC widens, opening the A-gate. RFC opens using a large hinge motion, pivoting around the B–C and C–D subunit interfaces (*Figure 3D, E*; *Video 3*). The RFC-E subunit uses its E-plug to bind PCNA, which pulls the A' domain and E-plug up to 45 Å away from the AAA+ module. This reveals a large opening of the A-gate (at its most narrow,

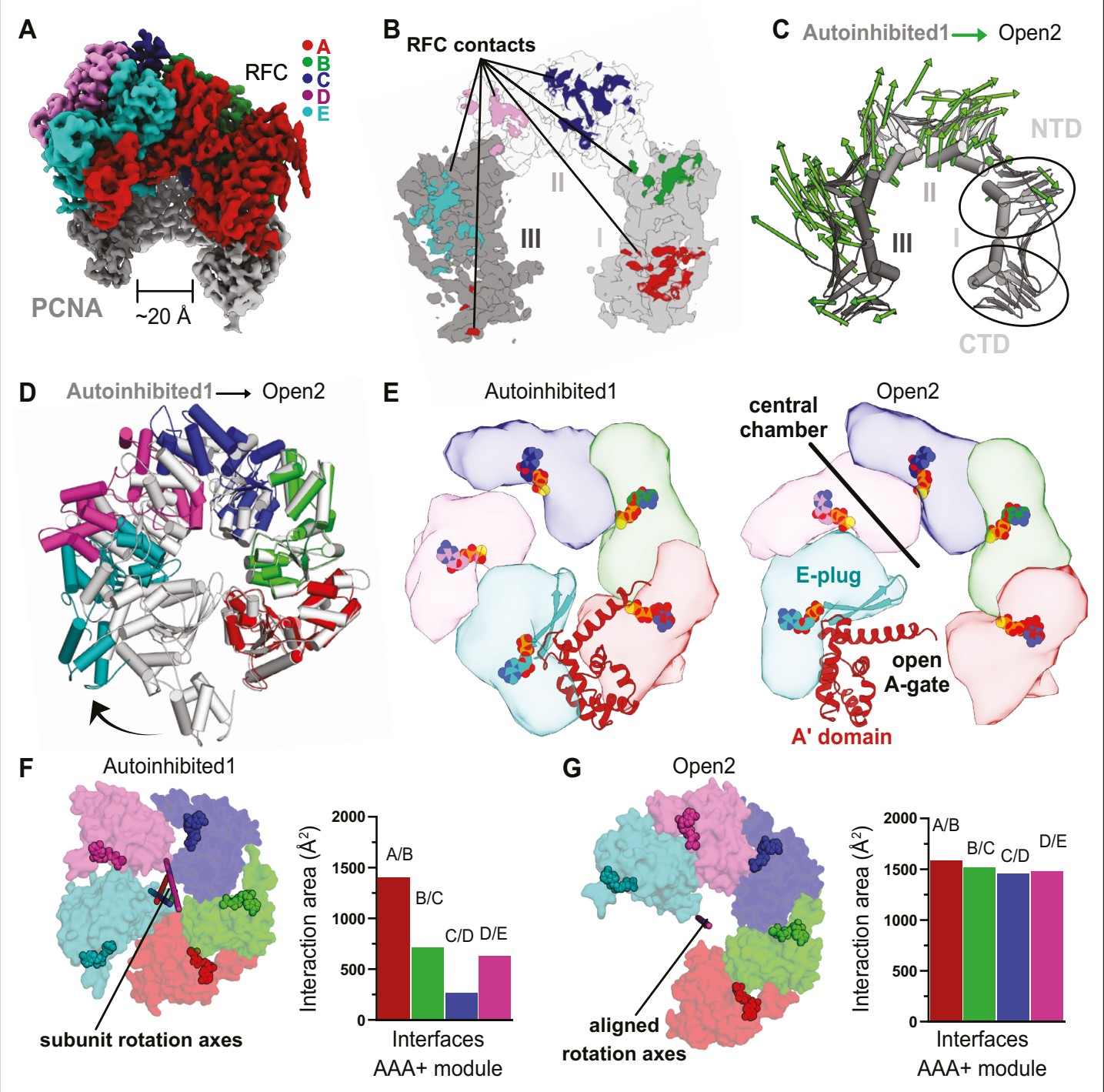

**Figure 3.** RFC undergoes a large conformational change to open PCNA. (**A**) Cryo-EM map of RFC bound to an open PCNA ring. (**B**) PCNA is held open by contacts with all five subunits of RFC. (**C**) The $C_\alpha$ displacement from closed to open PCNA is indicated by arrows, scaled up by a factor of 4. (**D**) The AAA+ modules widen from the Autoinhibited state (gray) to an open spiral conformation. (**E**) Top view of the AAA+ spiral shows that the E-plug and A-gate block access to RFC's central DNA-binding chamber in the Autoinhibited conformation but retract in the open conformation. RFC opens wide enough for DNA to directly enter the central chamber. (**F**) Top view of the Rossmann fold arrangement in the Autoinhibited conformation. The rotation axes that relate neighboring subunits are shown in different colors and are skewed, indicating asymmetric rotations which lead to gaps between the subunits. (**G**) The rotation axes overlay in the Open2 state of RFC, indicating a symmetric arrangement of the AAA+ spiral. Symmetrization closes the gaps, and results in an increased interaction area between neighboring subunits.

The online version of this article includes the following figure supplement(s) for figure 3:

*Figure 3 continued on next page*

**Figure supplement 1.** Comparison of RFC bound to open PCNA from different datasets.

**Figure supplement 2.** RFC interacts with all five subunits to hold PCNA open.

the A-gate is approximately 20 Å wide) (*Figure 3E*). p/t-DNA can therefore directly enter the open RFC:PCNA complex.

Opening of the A-gate separates the RFC-A Lid and collar domains, inducing a fold-switching transition in the Lid domain. The majority of the last helix of the Lid (Helix α4; residues 541–546) unravels into a taut β-strand conformation (*Figure 4A, B*). The remaining residues in helix α4 (residues 536–542) shift forward, causing a major change in the core packing of the Lid domain. This 'sliding spring' motion leads to a ~11 Å helix displacement, whereby some residues, such as Leu 549, move up to 22 Å from their original position. The stretching of the RFC-A Lid opens a new pore between the A and B subunits (*Figure 4B*). We discuss the role of this pore in the next section.

## Structures of the RFC:PCNA complex bound to primer–template DNA

To reveal how RFC:PCNA binds and responds to DNA, we analyzed two classes that contain DNA-bound RFC:PCNA. One class shows PCNA in an open lock-washer shape, and the other has PCNA in

**Table 3.** Cryo-EM data collection, processing, and model statistics.

| Dataset | No DNA | | | | | DNA | |
|---|---|---|---|---|---|---|---|
| Magnification | 130,000 | | | | | 81,000 | |
| Voltage (keV) | 300 | | | | | 300 | |
| Cumulative exposure (e−/Å 2) | 49–51 | | | | | 40 | |
| Detector | K2 Summit | | | | | K3 | |
| Pixel size (Å) | 1.059 | | | | | 1.06 | |
| Defocus range (µm) | −1.1 to −2.4 | | | | | −1.2 to −2.3 | |
| Micrographs used (no.) | 6109 | | | | | 4499 | |
| Initial particle images (no.) | 954,291 | | | | | 1,331,440 | |
| Symmetry | C | | | | | | |
| Class name | Autoinhibited1 | Autoinhibited2 | Autoinhibited3 | Open1 | Open2 | DNA-open | DNA-closed |
| Final refined particles (no.) | 55,308 | 68,227 | 60,036 | 46,069 | 63,752 | 46,300 | 76,270 |
| Applied B factor (Å²) | −100 | −159.352 | −163.938 | −100 | −106.457 | −105.857 | −105.313 |
| Map resolution (Å, FSC 0.143) | 3.8 | 3.9 | 4.0 | 4.0 | 3.5 | 3.4 | 3.3 |
| Model-Map CC_mask | 0.78 | 0.77 | 0.77 | 0.76 | 0.78 | 0.79 | 0.77 |
| Bond lengths (Å), angles (°) | 0.002, 0.585 | 0.002, 0.561 | 0.002, 0.558 | 0.002, 0.574 | 0.002, 0.542 | 0.002, 0.518 | 0.002, 0.523 |
| Ramachandran outliers, allowed, favored | 0.00, 3.16, 96.84 | 0.00, 3.11, 96.89 | 0.00, 2.89, 97.11 | 0.00, 3.08, 96.92 | 0.00, 3.38, 96.62 | 0.00, 2.23, 97.77 | 0.00, 2.16, 97.84 |
| Poor rotamers (%), MolProbity score, Clashscore (all atoms) | 0.00, 1.68, 9.05 | 0.00, 1.68, 9.42 | 0.00, 1.68, 9.95 | 0.00, 1.67, 9.26 | 2.01, 1.91, 8.67 | 1.09, 1.54, 8.44 | 1.09, 1.55, 9.18 |
| Accession number, EMDB, PDB | 25568, 7THJ | 25569, 7TIC | 25614, 7THV | 25615, 7TKU | 25753, 7TI8 | 25616, 7TIB | 25617, 7TID |

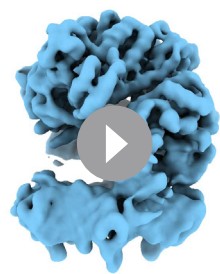

**Video 1.** RFC:PCNA motion along Eigenvalue 1 with masks on PCNA and RFC.
https://elifesciences.org/articles/74175/figures#video1

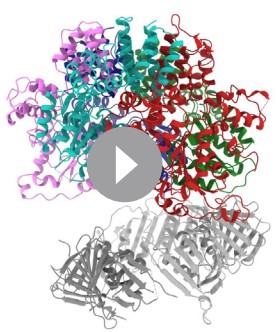

**Video 3.** Morph closed to open.
https://elifesciences.org/articles/74175/figures#video3

a closed conformation. Therefore, we term these two states DNA$^{PCNA-open}$ and DNA$^{PCNA-closed}$, respectively (*Figure 5A–C*). Both classes contain clear density for p/t-DNA: 18 basepairs of duplex DNA are bound inside the central chambers of RFC and PCNA, and 6 nucleotides of the ssDNA template extend through the A-gate, preventing its closure. The AAA+ spiral of RFC tracks the minor groove of dsDNA using a suite of residues that are conserved across all known clamp loaders to match the helical symmetry of DNA (*Kelch et al., 2011*; *Simonetta et al., 2009*).

The E-plug beta-hairpin slots into the major groove of the duplex region of p/t-DNA (*Figure 5D*). Conserved basic residues at the tip of the E-plug interact directly with both the template and primer strands. Therefore, the E-plug provides a mechanism for the RFC AAA+ spiral to recognize both strands of DNA, unlike the clamp loaders from *E. coli* and T4 phage, whose AAA+ spirals only interact with the template strand (*Kelch et al., 2011*; *Simonetta et al., 2009*). Moreover, this structure shows that the E-plug changes its role from blocking DNA binding (in the three Autoinhibited states) to one in which it directly supports DNA binding. This explains the nonintuitive effect on DNA binding we observed previously, where hRFC variants with a mutated E-plug bind DNA with equivalent affinity as WT-hRFC (*Gaubitz et al., 2020*).

In DNA$^{PCNA-open}$, both RFC and PCNA broadly resemble the conformations seen in Open1 and Open2. The RFC A-gate is open, with all five subunits gripping PCNA in an open lock-washer shape. However, both RFC and PCNA constrict relative to the Open1 and Open2 structures (*Figure 5G, H* and *Figure 5—figure supplement 1A, B*). RFC constricts modestly, pivoting the E, D, and C subunits around a hinge at the B–C interface. PCNA constricts ~12 Å upon DNA binding, with most of this constriction occurring in subunit III of PCNA (*Figure 5—figure supplement 1A*). Subunit III of PCNA is held by the RFC-D and RFC-E subunits, although RFC-E grips PCNA less tightly in DNA$^{PCNA-open}$ (~3800 Å$^2$ total RFC–PCNA interaction area for the Open1 and 2 structures vs ~3400 Å$^2$ for DNA$^{PCNA-open}$). Overall, the PCNA conformation is similar to that seen for the structure of the T4 phage clamp bound to clamp loader and p/t-DNA (*Kelch et al., 2011*).

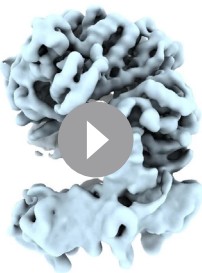

**Video 2.** RFC:PCNA motion along Eigenvalue 2 with masks on PCNA and RFC.
https://elifesciences.org/articles/74175/figures#video2

The DNA$^{PCNA-closed}$ structure has a closed PCNA ring that is distorted from planarity. Upon closure, PCNA loses its interaction with the RFC-E subunit, but retains its interfaces with the other four RFC subunits (*Figure 5E, F*). The distortion of the PCNA ring is most prevalent in subunit III, which puckers to maintain its interaction with the RFC-D subunit (*Figure 5—figure supplement 1A, B*). Interestingly, the interaction between DNA and PCNA becomes more extensive upon PCNA closure (~50 vs 250 Å$^2$). Conserved basic residues lining the inner pore of PCNA also interact directly with the duplex, as has been hypothesized

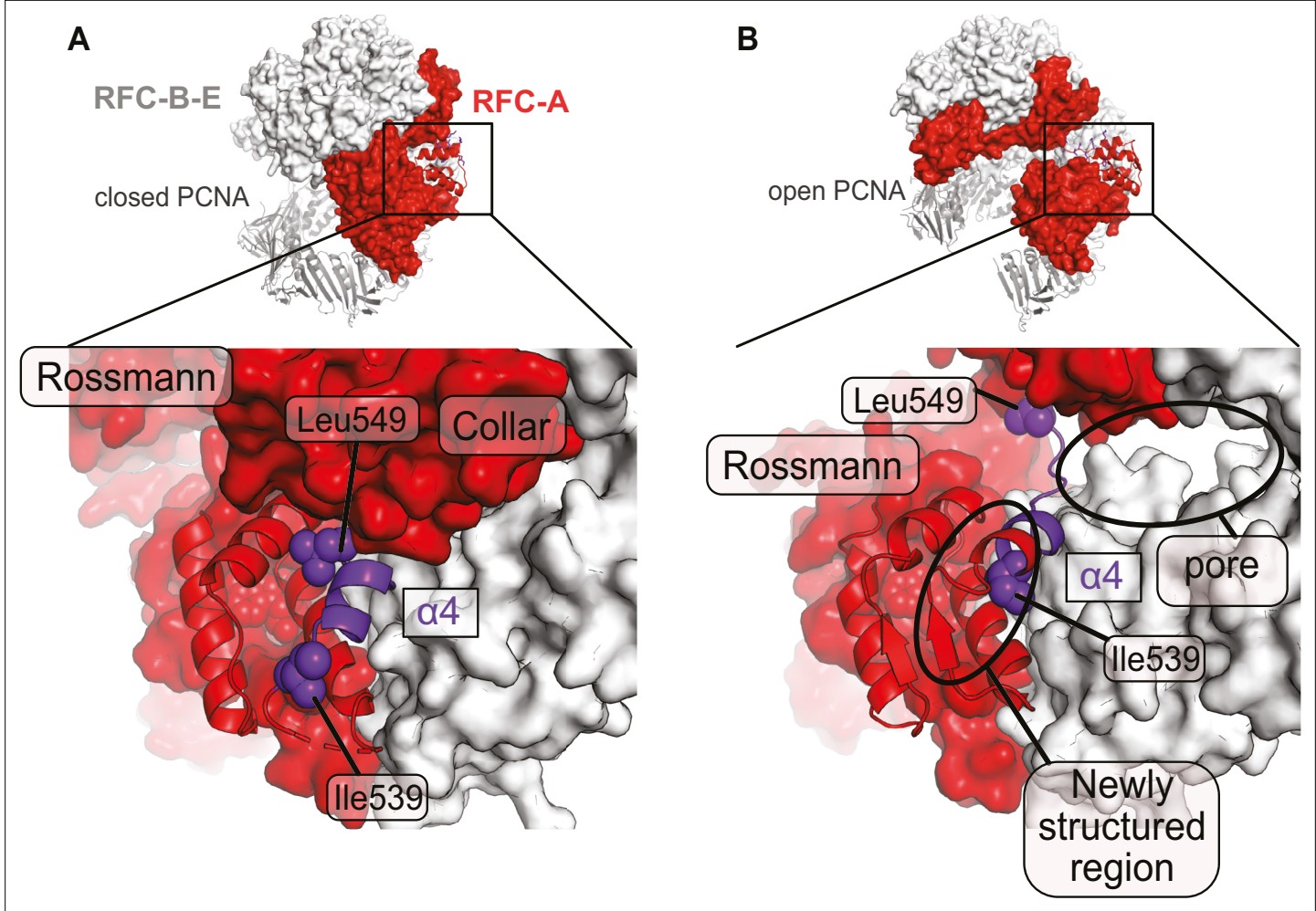

**Figure 4.** A fold-switch mechanism for opening a pore in the Open state of RFC:PCNA. (**A**) Helix 4 of the RFC-A subunit in Autoinhibited1 is shown in purple. (**B**) In the open conformation, Lid Helix four is displaced and partially unravels, whereby the packing arrangement of the hydrophobic core of the lid domain in RFC-A changes. Ile536 and Leu549 move ~13 and ~22 Å from their original position and a pore is formed between the RFC-A and RFC-B subunits.

previously (*Liu et al., 2017*; *McNally et al., 2010*). We propose that these interactions help to drive the closure of PCNA around DNA.

DNA$^{PCNA-open}$ and DNA$^{PCNA-closed}$, just like the other states described herein, are in the fully ATP-bound state: ATPγS in the active sites of the A, B, C, and D subunits, and ADP in the nucleotide-binding site of the E subunit (*Figure 5—figure supplement 1C*). Therefore, these structures represent reaction intermediates following DNA binding but preceding ATP hydrolysis. Upon binding DNA, the AAA + spiral constricts (*Figure 5H*), primarily due to a hinge-like motion at the interface between RFC-C and RFC-B. The AAA + spiral constricts around an axis coincident with the DNA axis. Subsequent PCNA closure further exaggerates the constriction of the RFC AAA+ spiral (*Figure 5I*). Despite these movements, the position of the arginine finger within the ATPase active site does not change substantially (*Figure 5—figure supplement 2*). Thus, DNA binding likely stimulates ATP hydrolysis through another mode of action. One such proposed mode is repositioning a conserved arginine known as the switch residue, which in turn would activate the Walker B glutamate (*Kelch et al., 2011*; *Kelch et al., 2012*). However, we find that this residue is not in the position that was previously predicted to stimulate hydrolysis. Despite this, the active sites appear to be in the fully active state, with all of the catalytic machinery poised to hydrolyze ATP. We discuss the ramifications of this observation on the allosteric activation of RFC below.

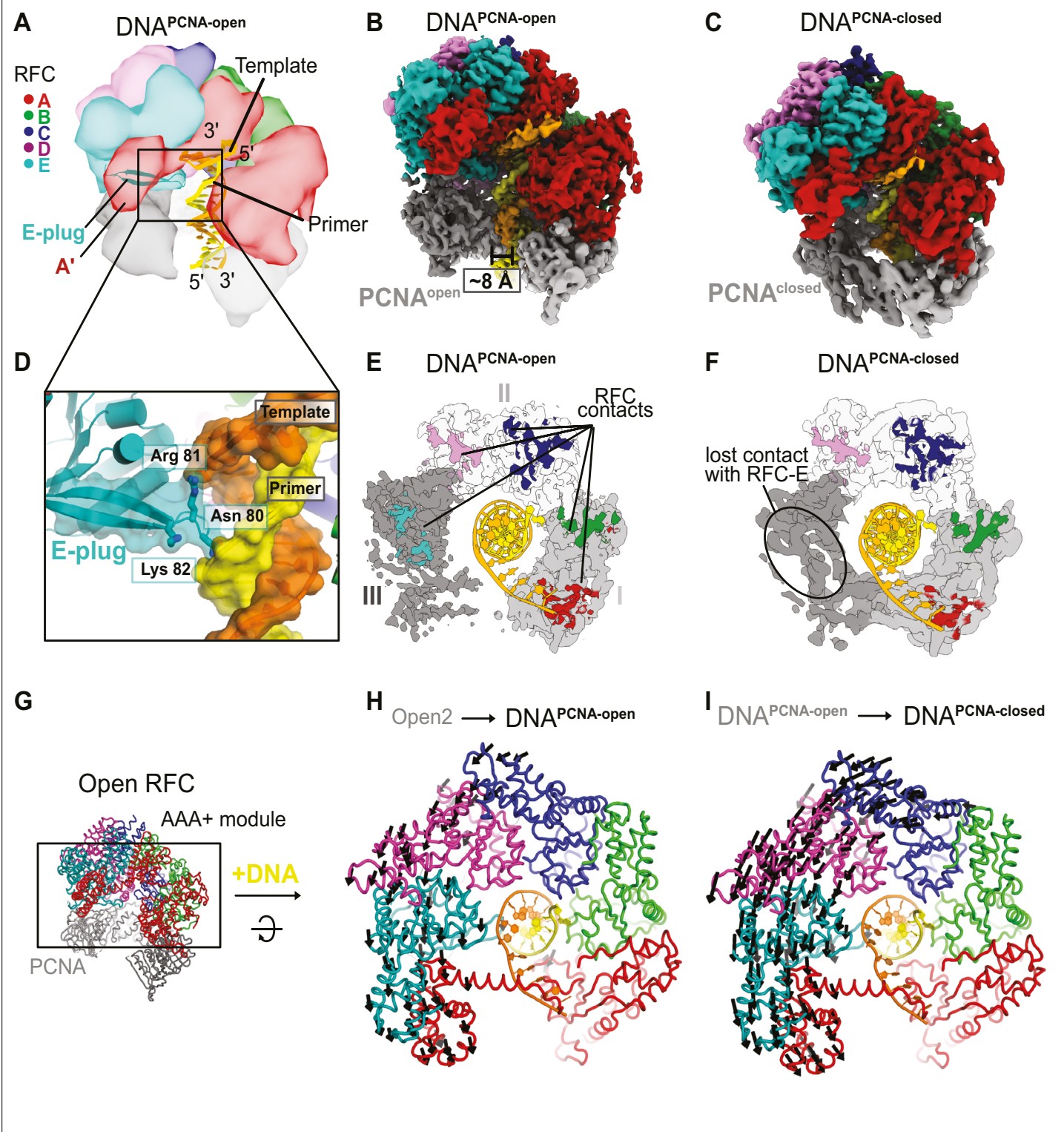

**Figure 5.** Structures of RFC:PCNA bound to DNA. (**A**) Schematic representation of the structure of RFC:PCNA bound to primer–template (p/t) DNA. (**B**) Cryo-EM map of RFC:PCNA bound to p/t-DNA and open PCNA (termed DNA$^{PCNA-open}$). (**C**) Cryo-EM map of RFC:PCNA bound to p/t-DNA with closed PCNA (termed DNA$^{PCNA-closed}$). (**D**) The E-plug inserts into the major groove and interacts with both strands of the p/t-DNA. (**E**) Top view of contact sites of RFC with PCNA. PCNA is held open by contacts with all five subunits in DNA$^{PCNA-open}$. (**F**) In DNA$^{PCNA-closed}$, the interaction between RFC-E and PCNA-III is lost.(**G**) Overview of structure of Open2. (**H**) Top view of the AAA+ spiral of DNA$^{PCNA-open}$. Displacement vectors between Open2 and DNA$^{PCNA-open}$

*Figure 5 continued on next page*

*Figure 5 continued*

are indicated by arrows, scaled up by a factor of 2. The AAA+ spiral constricts around DNA. (**I**) The AAA+ spiral of DNA$^{PCNA-closed}$. Displacement vectors between DNA$^{PCNA-open}$ and DNA$^{PCNA-closed}$ indicate that the AAA+ spiral constricts further around DNA, leading to changes in ATPase sites.

The online version of this article includes the following figure supplement(s) for figure 5:

**Figure supplement 1.** Conformational changes upon DNA binding and ring closure.

**Figure supplement 2.** Superposition of the ATPase active sites across conformations.

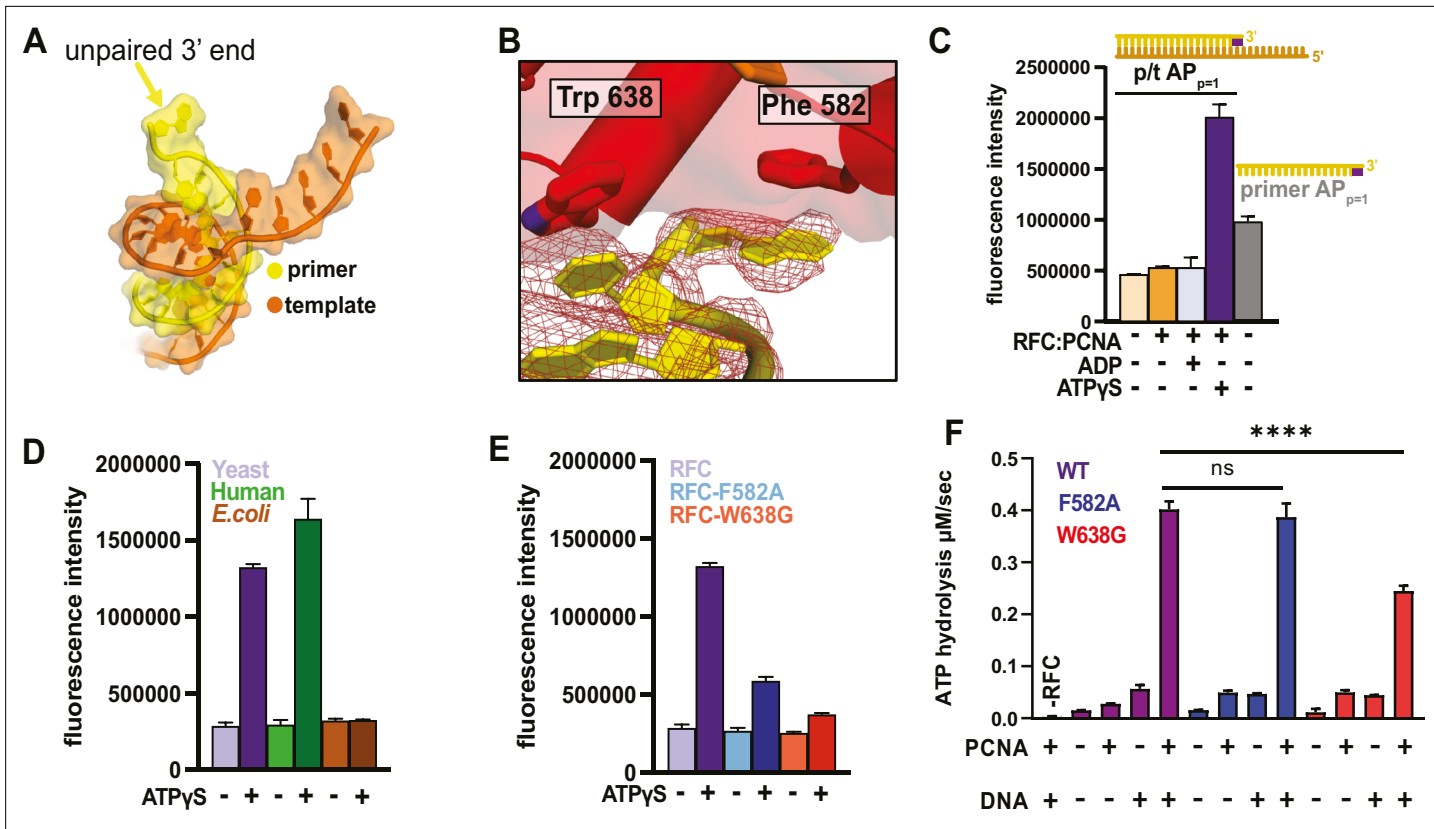

**Figure 6.** RFC separates the 3' end of the primer strand. (**A**) The last nucleotide in the primer strand is separated from the duplex. (**B**) The collar of RFC-A contains a 'separation pin' with two critical residues (Trp638 and Phe582) that stabilize the flipping of the 3' primer nucleotide into the pore between RFC-A and RFC-B. The cryo-EM map is shown in red mesh. (**C**) The primer strand of p/t-DNA contains 3' nucleotide with a 2-aminopurine (2AP) base, an adenine analog that reports on base-pairing and base-stacking. 2AP fluorescence increases in the presence of ATPγS and RFC:PCNA to a higher level than in the unpaired 2AP-labeled primer strand. (**D**) The human RFC:PCNA complex also induces an increase in 2AP fluorescence emission, whereas the *E. coli* clamp loader, which does not flip the 3' end of the primer (*Simonetta et al., 2009*), does not increase 2AP fluorescence. (**E**) Mutation of Phe582 and Trp638 reduces 2-AP fluorescence in the presence of ATPγS. (**F**) ATPase activity of the 'separation pin' mutants. The ATP hydrolysis rate of the RFC-W638G variant is significantly reduced compared to wild type in the presence of PCNA and DNA (p value from one-way ANOVA test: ****p ≤ 0.0001).

The online version of this article includes the following source data and figure supplement(s) for figure 6:

**Source data 1.** 2-Aminopurine fluorescence and ATPase data.

**Figure supplement 1.** The separation wedge has two critical residues.

**Figure supplement 1—source data 1.** 2-Aminopurine fluorescence and ATPase data.

**Figure supplement 2.** Discrimination of different primer–template junctions by RFC.

**Figure supplement 2—source data 1.** ATPase data.

**Figure supplement 3.** Functional characterization of separation pin mutants in *S. cerevisiae*.

**Figure supplement 4.** Differences in how duplex p/t-DNA is held in the central chamber of clamp loaders.

**Figure supplement 5.** Conservation of the separation pin.

## RFC flips the 3′ base of the primer strand

Unexpectedly, we observe that the 3′ nucleotide of the primer strand is melted in both DNA-bound RFC:PCNA structures, with the base flipped away from the rest of the duplex (*Figure 6A, B*). The basepair is disrupted by a 'separation pin' at the base of the RFC-A collar domain that wedges between the DNA strands (*Figure 6B*). The indole ring of Trp638 replaces the flipped 3′ base to maintain stacking interactions. The 3′ nucleotide is repositioned inside the pore formed by the unraveling of the RFC-A Lid domain upon opening of the A-gate; this site is closed in the Autoinhibited state (*Figure 4*). The flipped base stacks against the phenyl ring of Phe582. These residues are conserved in eukaryotic clamp loaders but are absent in bacterial, archaeal or phage clamp loaders (*Figure 6— figure supplement 5A*).

To characterize base-flipping, we measured binding of DNA substrates containing the adenine analog 2-aminopurine (2AP). Fluorescence of 2AP is dependent on base-pairing (*Frey et al., 1995*; *Jean and Hall, 2001*): fluorescence is low when 2AP is base paired, but high in the free state. To monitor melting, we placed 2AP either at the 3′ end of the primer strand ($AP_{p=1}$) or at the corresponding site in the template strand ($AP_{t=1}$). Importantly, we find a dramatic increase in 2AP fluorescence that is dependent on addition of RFC, PCNA and ATP analog (*Figure 6C*, *Figure 6—figure supplement 1A–C*). The increase in 2AP fluorescence is not observed in the presence of ADP, which does not support DNA binding (*Kelch et al., 2011*; *McNally et al., 2010*). Placement of 2AP at the p = 2 or p = 3 position of the primer yields diminished fluorescence, suggesting that only the 3′ base is flipped (*Figure 6—figure supplement 1B*). Therefore, our 2AP experiments validate that RFC- and PCNA-dependent 3′ end melting occurs in solution. The human clamp loader, which has a similar separation pin as yeast RFC (*Gaubitz et al., 2020*), greatly enhances 2AP fluorescence. However, the *E. coli* clamp loader, which binds p/t-DNA but does not melt the primer strand (*Simonetta et al., 2009*), does not alter fluorescence (*Figure 6D*). Thus melting of the 3′ nucleotide is a conserved activity of eukaryotic clamp loaders, but is likely not used by bacterial clamp loaders.

To determine the mechanism and role of primer melting, we modified the p/t-DNA and/or key residues in the separation pin and assessed their effects on base-flipping, ATP hydrolysis, and DNA affinity. The W638G and F582A variants have attenuated base-flipping as measured by 2AP fluorescence (*Figure 6E*, *Figure 6—figure supplement 1C*). However, DNA-dependent ATP hydrolysis is minimally affected, particularly in the F582A variant, whose ATPase rate and apparent affinity for DNA are similar to WT (*Figure 6F* and *Figure 6—figure supplement 1D, E*). These results indicate that base-flipping requires the separation pin, but base-flipping is not required for DNA binding or ATPase activation.

We hypothesized that the base-flipping mechanism functions to specifically recognize the 3′ end of the primer. By flipping the base, the separation pin could potentially act as a quality control mechanism to verify proper status of the primer end. We tested this hypothesis by measuring how WT-RFC and the W638G and F582A variants respond to various nucleic acid architectures. If our hypothesis were true, we would expect that the W638G and F582A variants would lose the ability to discriminate against 'incorrect' nucleic acid substrates. We tested ATPase activity against a series of nucleic acid substrates that include: ssDNA, 3′ phosphate, 3′ abasic sites, a 3′ ribonucleotide, an RNA primer, ssDNA–dsDNA junctions of opposite polarity (i.e. recessed 3′ ends, *Figure 6—figure supplement 2A*). We performed these assays using nucleic acid concentrations at or near the $K_d$ for the various forms of RFC (*Figure 6—figure supplement 1A*), so that any deviations in activity or binding would be observable. However, we observe nearly identical ATPase activity profiles for the variants as we do for WT-RFC (*Figure 6—figure supplement 2B–D*). Therefore, the biochemical characterization of variants with reduced base-flipping does not support our hypothesis that the separation pin acts to discriminate against incorrect substrates.

To directly assess the physiological role of base-flipping in normal RFC function, we measured growth of yeast strains carrying the WT, W638G, or F582A variants as the only copy of RFC1 (*Figure 6—figure supplement 3*). We tested yeast growth across a wide variety of DNA damaging treatments: ultraviolet radiation (UV), hydroxyurea (HU), or methyl methanesulfonate (MMS). Because base-flipping is thought to have a strong temperature dependence (*Yin et al., 2014*), we measured yeast growth over a broad temperature range (18–37°C). Surprisingly, we find no obvious growth phenotype across our broad spectrum of conditions (*Figure 6—figure supplement 3*). Thus, we currently find no obvious role for the separation pin, despite its conservation in RFC complexes across all eukaryotes. Further

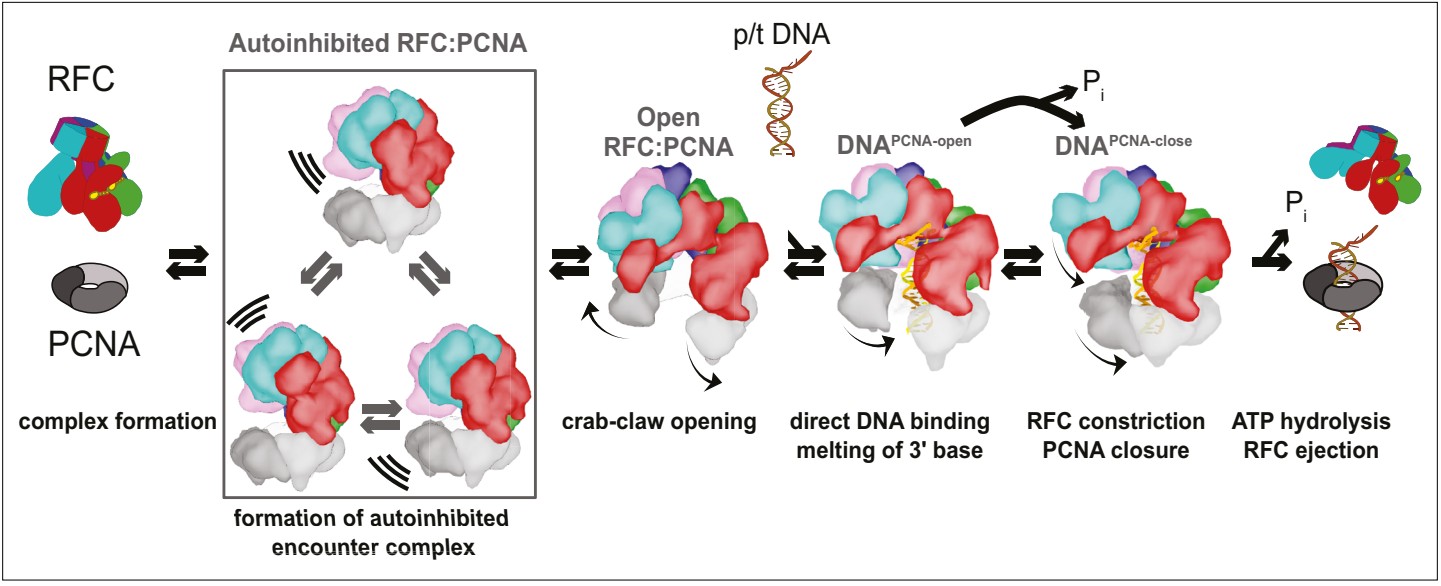

**Figure 7.** Clamp loading by RFC. Initial binding of RFC to PCNA places the complex in an Autoinhibited state, whereby closed PCNA and the E-plug preclude DNA binding, and an overtightened AAA+ helix inhibits ATPase activity. The Autoinhibited state is dynamic, rocking PCNA relative to RFC as captured by multibody refinement. Upon complete binding to PCNA, RFC uses the crab-claw mechanism to simultaneously open both PCNA and the A-gate, providing an entryway for p/t-DNA. p/t-DNA then binds directly through the A-gate and open PCNA, which are wide enough to accommodate dsDNA entry. The 3' end of the primer is flipped into the pore that is formed between RFC-A and RFC-B. PCNA closes to form additional contacts with DNA, partially detaching from RFC at the E subunit. Finally, ATPase activity and inorganic phosphate release eject RFC, leaving PCNA bound to p/t-DNA in the correct orientation.

investigation will be required to discern the functional role, if any, of the base-flipping mechanism of RFC.

## Discussion

### Defining the clamp loading reaction in high resolution

We have determined a series of structures that provide a high-resolution view of the clamp loading process. Our structures correspond to numerous reaction intermediates, allowing us to order the structures into a coherent description of the clamp loading reaction prior to ATP hydrolysis (*Figure 7* and *Video 4*). The Autoinhibited states represent the transient encounter complex that forms early in the clamp loading process before ring opening. The Open1 and 2 states represent the stable intermediate state in which PCNA is opened but DNA has yet to bind. The DNA$^{PCNA\text{-}open}$ structure contains p/t-DNA and an open clamp, which is the transient intermediate following DNA binding (*Liu et al., 2017*; *Marzahn et al., 2015*; *Sakato et al., 2012a*). Finally, the DNA$^{PCNA\text{-}closed}$ structure represents a possible stable intermediate that forms if ATP hydrolysis were stalled for whatever reason (*Marzahn et al., 2015*; *Sakato et al., 2012b*). Therefore, our structures delineate the conformational states that span the entire clamp opening and closing process, the central reaction of the clamp loading cycle.

### A crab-claw mechanism for opening the sliding clamp

Our structures show that RFC is in a constricted, autoinhibited conformation upon initial binding to PCNA. This state is highly dynamic, and we captured some of the conformational

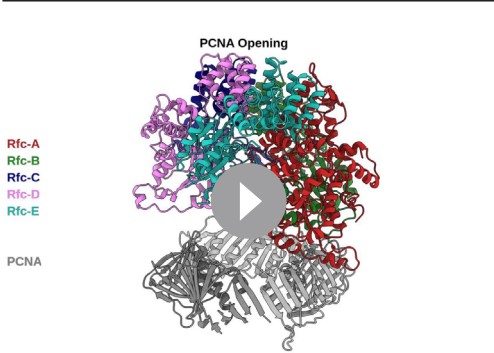

**Video 4.** PCNA loading by RFC.
https://elifesciences.org/articles/74175/figures#video4

heterogeneity using multibody refinement. The primary mode of motion pivots PCNA relative to RFC, such that PCNA approaches the D- and E-subunits of RFC. We speculate that this motion is on-pathway toward a direct interaction between PCNA and all five RFC subunits, facilitating the opening of the PCNA ring. Thus, the dynamics of the Autoinhibited complex are important for the opening of PCNA. Future studies will investigate this possibility.

To open PCNA, our structures show that RFC uses the previously hypothesized 'crab-claw' mechanism (*Jeruzalmi et al., 2001a*; *Jeruzalmi et al., 2001b*; *O'Donnell et al., 2001*). This contradicts the previous suggestion that the *E. coli* clamp loader opens the ring with limited conformational changes in the clamp loader (*Goedken et al., 2004*; *Kelch, 2016*). In this 'limited change' model, ATP binding places the encounter complex in a conformation that 'templates' the open clamp. However, our structures preclude this model for RFC because we observe large conformational changes in the clamp loader upon opening the PCNA ring. Furthermore, the Autoinhibited state of RFC cannot template an open PCNA conformation. One possible reason for the discrepancy between the two studies is that different model systems were used; bacterial clamp loaders lack the A' domain that constricts the AAA+ spiral of the yeast clamp loader. Without the A' domain, the bacterial clamp loaders may be free to adopt a conformation that can template the open clamp prior to clamp binding.

The crab-claw motion that we observe is primarily driven by a hinge-like motion that pivots about the RFC-C subunit. This motion allows the A' domain and E, D, and C subunits to grip PCNA tightly, which is impossible in the Autoinhibited state. Kinetic characterization of RFC variants has predicted a hinge role for this region (*Sakato et al., 2012b*), highlighting this subunit's importance in clamp loading. The crab-claw conformational change is remarkable because it requires a fold-switching event in the Lid domain of the RFC-A subunit (*Figure 4A, B*). At a minimum, this would require that helix-4 of the RFC-A Lid to unfold and refold into a new position. The fact that clamp opening is relatively fast (*Liu et al., 2017*) and does not require ATP hydrolysis indicates that these conformational rearrangements must have a relatively low energy barrier despite the large-scale motion. How the RFC:PCNA complex couples these motions becomes an important question for future studies.

Why use a 'crab-claw' mechanism? We envision two nonmutually exclusive hypotheses. First, we hypothesize that this mechanism allows RFC to bind each of its macromolecular substrates (PCNA and p/t-DNA) in the proper order to ensure efficient clamp loading and to avoid futile cycles of ATP hydrolysis. For proper clamp loading, RFC must bind PCNA first, because initial binding of p/t-DNA would sterically hinder binding of the PCNA ring. Therefore, RFC has evolved high affinity for PCNA and only binds p/t-DNA with high affinity after it has bound PCNA (*Cai et al., 1998*; *Shiomi et al., 2000*). The crab-claw mechanism for PCNA opening can explain this hierarchy of binding, as the autoinhibited state blocks the DNA-binding site (*Gaubitz et al., 2020* and *Figure 3E*). The crab-claw mechanism ensures that RFC's DNA-binding chamber only becomes accessible once the PCNA ring is open. Our second hypothesis is that the crab-claw mechanism enables complex modes of clamp loader regulation. Clamp loader activity could be inhibited by binding partners or post-translational modifications that favor the Autoinhibited state. There are numerous RFC binding partners and post-translational modifications that remain unexplored, and thus are candidates for playing regulatory roles (*Dephoure et al., 2008*; *Kim and Brill, 2001*; *Ochoa et al., 2020*; *Olsen et al., 2010*; *Tomida et al., 2008*; *Wang et al., 2012*; *Wang et al., 2013*).

## RFC-A subunit drives DNA recognition

To illuminate how RFC recognizes DNA, we measured the relative contribution of each RFC subunit to DNA binding. We find that RFC-A accounts for ~64% of the buried surface area between RFC and DNA. This contrasts with T4 and *E. coli* clamp loaders, where the A subunits account for ~36% of the binding interface (*Figure 6—figure supplement 4B*). Much of this proportional increase arises from additional interactions between RFC-A and DNA through the separation pin and the flipped 3' nucleotide. Furthermore, we find that B, C, D, and E subunits of RFC interact with DNA significantly less (~760 Å$^2$) than the comparable subunits of T4 and *E. coli* (~1125 Å$^2$). The decrease in DNA interaction from the B, C, D, and E subunits is due to the p/t-DNA duplex region inserting deeper into the AAA+ spiral of the T4 and *E. coli* clamp loaders than in RFC (*Figure 6—figure supplement 4A*). Therefore, the large swing in the proportional interaction area is the net result of additional interactions from RFC-A and less from the remaining subunits.

This proportionally large interaction area suggests RFC-A as the subunit primarily responsible for recognizing DNA. This finding provides an attractive explanation for how alternative clamp loaders specifically recognize different DNA structures. RFC-like complexes or RLCs are only found in eukaryotes and share four of RFC's five subunits (RFC-B through RFC-E); each RLC contains a unique A subunit (*Majka and Burgers, 2004*). We hypothesize that the diminished role of the B, C, D, and E subunits in DNA recognition allows the A-subunit to assume the role of specifically binding unique structures of DNA. In support of this hypothesis, bacterial and phage clamp loaders do not have alternative forms that recognize different DNA structures, and their clamp loaders have substantially more contact between DNA and the B, C, D, and E subunits. The more pronounced role of the A subunit in eukaryotic clamp loaders allows for dramatically more plasticity in function. Further, the diminished role of the remaining subunits raises the question of how the pivot point at the C subunit contributes to the activity of RLC complexes. Finally, these findings raise the intriguing possibility of engineering RLCs with novel specificity and activity.

Following this reasoning even further, we hypothesized that RFC flips the 3′ nucleotide to specifically recognize the recessed 3′ end of p/t-DNA. We observe flipping of the 3′ nucleotide in both the DNA$^{PCNA-open}$ and DNA$^{PCNA-closed}$ structures, indicating that flipping can occur before ring closure. This observation can explain the 'DNA repositioning transition' that occurs quickly ($t_{1/2}$ ~ 35 ms) after initial DNA binding, but before clamp closure (*Liu et al., 2017*). We propose that this transition is the flipping of the 3′ nucleotide. However, the flipping mechanism does not appear to be used to discriminate between different DNA architectures. The W638G and F582A variants have a similar DNA discrimination profile as WT-RFC, despite having very different base-flipping activity (*Figure 6E*, *Figure 6—figure supplement 2*). Moreover, the physiological role of base-flipping is unclear, as yeast carrying these variants have no obvious cellular defects (*Figure 6—figure supplement 3*). We still hypothesize that there is likely a role for the flipping activity, as the separation pin is conserved across RFC complexes from yeast to humans. Moreover, this separation pin is not found in the related 9-1-1 clamp loader Rad24-RLC (*Castaneda et al., 2021*; *Zheng et al., 2021*). A separation pin extension is found in the related loader Ctf18 (*Figure 6—figure supplement 5B*) but the flipping amino acids are not conserved. (Predictions for or against a separation pin in the final loader subunit Elg1 are weak due to very limited sequence homology between RFC1 and Elg1.) Future experiments will investigate the role of base-flipping in more detail.

## Forces driving clamp loading

Our structures delineate a conformational pathway that illustrates much of the clamp loading reaction. We reveal how: (1) RFC initially binds PCNA, (2) how PCNA is opened, (3) how DNA is bound, and (4) how PCNA closes around DNA. This unprecedented view into the mechanism of clamp loading allows us to hypothesize on the forces that drive this reaction toward the loading of PCNA. We use the interaction areas between and within PCNA, RFC, and p/t-DNA to approximate these forces.

PCNA is opened through a large conformational change in both PCNA and RFC. In solution the open form is the predominant state (*Zhuang et al., 2006*), so it is important to understand what interactions drive this opening. Upon opening, PCNA loses the entire interface between subunits I and III. However, the open PCNA ring increases its interaction area with RFC by contacting all five subunits. Moreover, the crab-claw motion of RFC results in tighter association between adjacent AAA+ modules. Altogether, the opening of PCNA and RFC result in an increased interaction area of ~4000 Å$^2$ (*Figure 3F*). We propose that this is the driving force for stabilizing the open form of PCNA.

Once open, p/t-DNA enters the PCNA:RFC complex through the A-gate. The A-gate is wide enough for dsDNA to directly enter into the RFC:PCNA inner chamber. This finding is in direct contradiction of the 'filter-and-slide' model for DNA binding that posited that the opening is large enough for only ssDNA to enter such that the clamp loader filtered out dsDNA to accelerate the search for a p/t-junction (*Kelch, 2016*; *Kelch et al., 2011*). The filter-and-slide model was primarily predicated on crystal structures of the T4 phage clamp loader and on FRET data that suggested that initial binding of DNA does not constrict the open clamp (*Kelch et al., 2011*; *Zhuang et al., 2006*). While it remains a possibility that other clamp loaders use a filter-and-slide mechanism, our structures clearly indicate that yeast RFC uses the much more simple direct binding model.

Once DNA is bound, PCNA must close around the ring before ejection of the RFC complex. Rapid kinetics studies showed that ATP hydrolysis precedes clamp closure under normal conditions

(*Liu et al., 2017*; *Marzahn et al., 2015*; *Sakato et al., 2012b*). Taken together, these two points could lead to the conclusion that ATP hydrolysis provides the energy to actively close the clamp after loading DNA. However, we observe this transition from our DNA[PCNA-open] and DNA[PCNA-closed] structures, and neither structure shows evidence of ATP hydrolysis, suggesting that PCNA can close before ATP hydrolyzes. To harmonize all of the available data, we must draw a new conclusion, which is that while ATP hydrolysis typically occurs prior to clamp closure, it is not strictly required, and clamp closure can precede hydrolysis if the hydrolysis step becomes rate limiting, as would likely occur with the slowly hydrolyzable ATPγS. It still remains possible that ATP hydrolysis could make clamp closure *easier*, by weakening interactions between RFC and PCNA/DNA, but in this view clamp closure is still a spontaneous process and does not require harvesting energy from ATP hydrolysis. Therefore, it is possible that ATP hydrolysis can proceed from either DNA[PCNA-open] and DNA[PCNA-closed] states, but most commonly from the DNA[PCNA-open] state.

This raises the question as to how DNA stimulates ATP hydrolysis and subsequent ejection of the clamp loader. We note that the ATPase active sites do not change much from the Open to DNA[PCNA-open] or DNA[PCNA-closed] conformations (*Figure 5—figure supplement 2*). It is also surprising that the AAA+ modules are already in a symmetrized pose prior to DNA loading, because DNA had been thought to be the driving force for symmetrizing the AAA+ spiral (*Kelch et al., 2011*; *Simonetta et al., 2009*), and this symmetry had been thought to favor ATP hydrolysis. Despite this symmetry, the RFC:PCNA complex (corresponding to the Open1 and Open2 structures) has ~five- to tenfold lower ATPase activity than when both PCNA and DNA are bound (*Figure 6F*, *Chen et al., 2009*; *Gomes et al., 2001*; *McNally et al., 2010*; *Sakato et al., 2012a*). This implies that, whereas clamp opening is both necessary and sufficient for symmetrizing the AAA+ modules, this symmetry by itself is not sufficient to stimulate ATP hydrolysis.

There remain many possible avenues for DNA to stimulate ATP hydrolysis. In many AAA+ enzymes, it has been shown that certain residues couple ligand binding and ATP hydrolysis by activating the Walker B glutamate residue (*Zhang and Wigley, 2008*). A set of conserved arginines (termed the arginine switch residues) within the core of the AAA+ module were proposed to play this role in clamp loaders (*Kelch et al., 2011*). The arginine switch residues had been hypothesized to flip outward to grip DNA, thereby releasing the Walker B catalytic glutamate to activate ATP hydrolysis. However, the lack of flipping of the arginine switch residues in response to DNA binding in our structures argues that the proposed arginine switch mechanism is not critical for sensing and responding to DNA binding. Our observations are in agreement with previous studies that found that the arginine switch residues of RFC do not likely play a direct role in activating ATP hydrolysis, but are important for the synergistic activation by both PCNA- and DNA binding (*Liu et al., 2017*). An alternative route, involving a different arginine residue interacting with the ATPase active site, has recently been proposed for DnaC and extended to RFC (*Puri et al., 2021*). However, we again do not see structural evidence supporting this mechanism. We cannot rule out these mechanisms (or a combination of the two), as these types of interactions may occur just before hydrolysis and are not readily apparent in stalled structures. A recent study on the T4 clamp loader suggests that structural rigidity of a 'central coupler' that encircles DNA is important for hydrolysis (*Subramanian et al., 2021*). Thus, tight binding of RFC to DNA could provide rigidity necessary to stimulate ATP hydrolysis.

Lastly, we note that while DNA[PCNA-open] and DNA[PCNA-closed] have similar overall interaction areas, PCNA interacts with DNA much more intimately in the DNA[PCNA-closed] structure, with direct contact to several conserved basic residues lining the PCNA inner pore. Lys20, Arg80, and Arg147 in particular show close interaction with the PCNA ring. These residues have been independently identified as critical for efficient DNA binding, ATP hydrolysis, and clamp loading (*McNally et al., 2010*; *Zhou and Hingorani, 2012*). Therefore, PCNA is an allosteric effector in its own loading and its role in stimulating ATPase activity upon DNA binding should not be overlooked. Further studies will be necessary to reveal how RFC integrates binding of both PCNA and p/t-DNA to achieve full activation.

## Comparison with other AAA+ machines

Clamp loaders have long been models for structure and mechanism of AAA+ proteins (*Guenther et al., 1997*). However, they are unusual in that they are pentameric protein remodeling switches instead of the more typical hexameric rings that act as processive motors (*Hanson and Whiteheart, 2005*; *Kelch, 2016*). We note that conformational changes that we observe here in RFC appear to be

more dramatic than those typically seen during motor function. This is likely because the constraints imposed by ring closure limits the types of motions that are available. On the other hand, the open nature of the RFC complex is less constrained and so can adopt more dramatic conformational changes. We further note that these types of large conformational changes are more commonplace in other members of the Initiator/Loader class of AAA+ machines. We propose that the open nature of this class provides larger conformational variability that is necessary for the regulation of these switch-like machines.

# Materials and methods

## Key resources table

| Reagent type (species) or resource | Designation | Source or reference | Identifiers | Additional information |
|---|---|---|---|---|
| Strain, strain background (*Escherichia coli*) | BL21(DE3) | Novagen | 69,450 | Chemically competent cells |
| Recombinant DNA reagent | pET(11a)-RFC[2 + 3 + 4] (plasmid) | *Finkelstein et al., 2003* | | Expression plasmid |
| Recombinant DNA reagent | pLANT-2/RIL[1 + 5] (plasmid) | *Finkelstein et al., 2003* | | Expression plasmid |
| Recombinant DNA reagent | pRS413-RFC1 (plasmid) | This study | | Plasmid for yeast expression of Rfc1 from endogenous promotor |
| Strain, strain background (*S. cerevisiae*) | BY4743 his3Δ1/his3Δ1 leu2Δ0/leu2Δ0 LYS2/lys2Δ0 met15Δ0/MET15 ura3Δ0/ura3Δ0 Δrfc1::KanMX4/RFC1 (YOR217W) | Dharmacon | YSC1055 (22473) | Yeast Heterozygous Collection |
| Software, algorithm | RELION | doi:10.7554/eLife.42166 | Relion 3.0.2 | |
| Software, algorithm | cisTEM | doi:10.7554/eLife.35383 | cisTEM-1.0.0-beta | https://cistem.org/software |
| Software, algorithm | Ctffind | doi:10.1016 /j.jsb.2015.08.008 | Ctffind 4.1 | |
| Software, algorithm | UCSF Chimera | UCSF, doi:10.1002/jcc.20084 | | http://plato.cgl.ucsf.edu/chimera/ |
| Software, algorithm | ChimeraX | UCSF, doi:10.1002/pro.3943 | ChimeraX-1.2 | https://www.cgl.ucsf.edu/chimerax/ |
| Software, algorithm | COOT | doi:10.1107/S0907444910007493 | Coot-0.9.4 | http://www2.mrc-lmb.cam.ac.uk/personal/pemsley/coot/ |
| Software, algorithm | Phenix | doi:10.1107/S0907444909052925 | Phenix-dev-3699 | https://phenix-online.org |
| Software, algorithm | PyMOL | PyMOL Molecular Graphics System, Schrodinger LLC | | https://www.pymol.org/ |
| Software, algorithm | GraphPad PRISM | GraphPad | GraphPad PRISM 9.2.1 | http://www.graphpad.com/ |
| Other | Pyruvate kinase | Calzyme | 107A0250 | |
| Other | Lactate Dehydrogenase | Worthington Biochemical Cooperation | LS002755 | |
| Other | Phosphoenol-pyruvic acid monopotassium salt | Alfa Aesar | B20358 | |

## Protein purification

RFC was purified as described previously with minor modifications (*Finkelstein et al., 2003*). pET(11a)-RFC[2 + 3 + 4] and pLANT-2/RIL-RFC[1 + 5] were transformed into BL21(DE3) *E. coli* cells (Millipore). After preculture, transformants were grown in 4 l of prewarmed terrific broth medium supplemented with 50 µg/ml kanamycin and 100 µg/ml ampicillin at 37°C and induced with IPTG at an optical density of 0.8. Protein expression was continued at 18°C for 15 hr. Cells were pelleted and resuspended in 300 ml lysis buffer (30 mM 2-[4-(2-hydroxyethyl)piperazin-1-yl]ethanesulfonic acid (HEPES)–NaOH pH 7.5, 250 mM NaCl, 0.25 mM Ethylenediaminetetraacetic acid (EDTA), 5% glycerol, 2 mM Dithiothreitol (DTT), 2 µg/ml aprotinin, 0.2 µg/ml pepstatin, 2 µg/ml leupeptin, and 1 mM phenylmethylsulfonyl fluoride (PMSF)). RFC was purified by chromatography over a 10 ml SP-Sepharose column (80 ml gradient of 300–600 mM NaCl in Buffer C) and a 10 ml Q-Sepharose column (40 ml gradient of 150–500 mM NaCl in Buffer C, GEHealthcare). Peak fractions of hRFC were pooled and dialyzed overnight into a buffer with 30 mM HEPES–NaOH pH 7.5, 250 mM NaCl, 5% glycerol, and 2 mM DTT.

PCNA was purified as described previously with modifications (*McNally et al., 2010*). BL21(DE3) *E. coli* cells were transformed with a pET-28 vector that encodes PCNA with a Precission protease cleavable N-terminal 6-His tag. After transformation, preculture and induction, 1 l of cells was grown overnight at 18°C in terrific broth medium supplemented with 50 µg/ml kanamycin. Cells were pelleted and resuspended 30 mM HEPES, pH 7.6, 20 mM imidazole, 500 mM NaCl, 10% glycerol, and 5 mM b-mercaptoethanol. The cells were lysed, centrifuged, and the filtered lysate was applied to a 5 ml HisTrap FF column (GE Healthcare). The column was washed with a buffer at 1 M NaCl, and subsequently washed with a buffer at a low salt concentration (50 mM NaCl). PCNA was eluted with a step of 50% with 500 mM imidazole. The eluted protein was cleaved with Precission protease for 2 hr at room temperature and applied to a 5 ml HiTrap Q HP column (GE Healthcare). Protein was eluted from the Q HP column with a 2 M NaCl buffer in a 100 ml gradient. Peak fractions were dialyzed against buffer containing 30 mM Tris, pH 7.5, 100 mM NaCl, and 2 mM DTT. Purified proteins were concentrated with an Amicon concentration device, aliquoted and frozen in liquid nitrogen for storage at −80°C.

## Crosslinking and mass spectrometry

RFC and PCNA were mixed in a 1/1 ratio and gel filtered into 1 mM tris(2-carboxyethyl)phosphine (TCEP), 200 mM NaCl, 25 mM HEPES–NaOH, pH 7.5, and 4 mM $MgCl_2$. The protein complex was diluted to 3 µM and after the addition of 1 mM ATPγS and 3-min incubation, 1 mM of bis(sulfosuccinimidyl)suberate (BS3, Thermo Scientific Pierce) was added for crosslinking. For crosslinking of DNA-bound RFC:PCNA, 1 mM ATPγS was added to the protein complex first and incubated for 2 min. 7 µM primer/template DNA was added and incubated for another 1 min. The primer sequence was 5′-GCAGACACTACGAGTACATA-3′ and the template sequence was 5′-TTTTTTTTTTTATGTACTCGTAGT GTCTGC-3′. Crosslinking was started with 1 mM BS3, incubated for 15 min at room temperature, and neutralized with Tris–HCl.

Sample without DNA was analyzed by mass spectrometry. The sample was reduced, alkylated, and loaded onto sodium dodecyl sulfate–polyacrylamide gel electrophoresis (SDS–PAGE gel). The gel band corresponding to the crosslinked complex >150 kDa was excised, destained, and incubated with trypsin. The digested peptides were extracted and desalted as previously described (*Peled et al., 2018*) and analyzed with LC–MS coupled to a Thermo Fisher Scientific Q Exactive Mass Spectrometer in data-dependent mode selecting only precursors of 3. The data were searched against the UniProt database, using Byonic and XlinkX of the Proteome Discoverer 2.3 package.

## Electron microscopy

### Negative-staining EM

100 nM of RFC:PCNA was applied on carbon-coated 400-mesh grids. Excess sample was blotted from the grid surface, the grids were washed twice with 50 mM HEPES, pH 7.5 and stained with 1% uranyl acetate. RFC:PCNA was imaged on a 120 kV Philips CM-120 microscope fitted with a Gatan Orius SC1000 detector.

### Cryo-EM sample preparation

Quantifoil R 0.6/1 (DNA dataset) grids were washed with ethyl acetate. Quantifoil and C-flat grids (Electron Microscopy Sciences) were glow discharged with Pelco easiGlow for 60 s at 25 mA (negative polarity). 2.8–3 µl sample was applied to grids at 10°C and 95% humidity in a Vitrobot Mark IV (FEI). Samples were blotted with a force of 5 for 5 s after a 2 s wait and plunged into liquid ethane.

### Cryo-EM data collection

RFC:PCNA was imaged on a Titan Krios operated at 300 kV and equipped with an GIF energy filter at ×130,000 magnification and a pixel size of 0.53 Å using a K2 Summit detector in superresolution counting mode. The data were collected in four sessions with a target defocus range of −1.1 to −2.4 and a total exposure of ~49–51 e−/Å$^2$ per micrograph averaging 50 frames. Image shift was used to record three images per hole with SerialEM (*Mastronarde, 2003*). Defective micrographs were discarded leaving a total of 6109 micrographs for processing. RFC:PCNA:DNA was imaged on a Titan Krios operated at 300 kV at ×81,000 magnification and a pixel size of 0.53 Å with a K3 detector in super-resolution mode. A total of 4499 micrographs were collected in 1 day with a target defocus of −1.2 to −2.3 and a total exposure of ~40 e−/Å$^2$ per micrograph averaging 30 frames.

## Data processing

Micrograph frames were aligned in IMOD (*Kremer et al., 1996*) with 2× binning, resulting in a pixel size of 1.06 Å/pixel. Initial CTF estimation and particle picking were performed using cisTEM (*Grant et al., 2018*; *Rohou and Grigorieff, 2015*). Following particle picking, particles were extracted with a box size of 240 pixels and subjected to 2D classification into 100 classes. Particles from classes with well-defined features were selected for processing in Relion (*Figure 1—figure supplement 2A, B*, *Figure 3A, B*). Coordinates and combined micrographs were imported into Relion 3.0.2 (*Zivanov et al., 2018*), CTF parameters were re-estimated with Gctf1.06 (*Zhang, 2016*) and particles were subjected to several rounds of 3D classification (*Figure 1—figure supplements 2D and 3C*). For 3D classification of the RFC:PCNA dataset, an ab initio model was generated with cisTEM, downfiltered to 50 Å and used as reference (*Figure 1—figure supplement 2C*). For 3D classification of the RFC:PCNA:DNA dataset, class Open1 of the RFC:PCNA dataset was downfiltered to 60 Å and used as reference. Selected, well resolved 3D classes were refined with Relion. The cryo-EM density was postprocessed in Relion for estimating the resolution and density modified with PHENIX for model building and refinement (*Terwilliger et al., 2020 Table 3*). Model information was not used during density modification.

## Model building and refinement

The crystal structure of yeast RFC bound to PCNA (PDB ID: 1SXJ) was used for initial fitting of Autoinhibited1. All subunits were split into globular domains and fitted into the cryo-EM density with UCSF Chimera (*Pettersen et al., 2004*). The model was adjusted in Coot (*Emsley and Cowtan, 2004*), and real-space iteratively refined with two macrocycles in PHENIX1.17 (*Liebschner et al., 2019*). Autoinhibited2,3 cryo-EM densities were rigid body fit with the refined model of Autoinhibited1, manually adjusted in coot and refined.

The refined model of Autoinhibited1 (*Figure 1—figure supplement 1C*) was fragmented into individual subunit domains and rigid body fitted into the cryo-EM density of Open2. The resulting model was further flexibly fitted and refined with Namdinator (*Kidmose et al., 2019*). The resulting model was adjusted in Coot, and refined in PHENIX. The model of Open2 was used for fitting the Open1 cryo-EM density. The fitted model was manually adjusted in Coot and refined in PHENIX. The cryo-EM density of DNA$^{PCNA-closed}$ (*Figure 1—figure supplement 3C*) was fitted using the Autoinhibited1 model and DNA was modeled in Coot. The resulting model was further flexibly fitted and refined with Namdinator (*Kidmose et al., 2019*). The model was then adjusted in Coot, and refined in PHENIX. The Namdinator output model of DNA$^{PCNA-closed}$ was used for fitting of the DNA$^{PCNA-open}$ cryo-EM density. The fitted model was manually adjusted in Coot and subjected to refinement in PHENIX. Interface areas were analyzed with the PISA server (*Krissinel and Henrick, 2007*). UCSF Chimera and Pymol were used for figure generation (*Delano, 2002*; *Pettersen et al., 2004*).

## ATPase assays

0.3 µM (*Figure 6F*) or 0.15 µM RFC (*Figure 6—figure supplement 1D*) was incubated with a master mix (3 U/ml Pyruvate kinase, 3 U/ml lactate dehydrogenase, 1 mM ATP, 670 µM phosphoenol pyruvate, 170 µM NADH, 50 mM Tris (pH 7.5), 0.5 mM TCEP, 5 mM MgCl$_2$, 200 mM potassium glutamate, 40 mM NaCl), 1 µM PCNA, and annealed primer/template DNA (2 µM *Figure 6F*, varying amounts *Figure 6—figure supplement 1D*). ATPase activity was measured at 25°C with the 2014 EnVison Multilabel Plate Reader to detect NAD+. Rates were obtained from a linear fit of the slopes using GraphPad Prism. For the ATPase activity measurements shown in *Figure 6—figure supplement 2*, 0.12 µM RFC was incubated with 1 µM PCNA and 0.03 µM different DNA constructs (as described in *Table 4*) and the master mix and buffer described above. ATPase activity was measured at room temperature. For each data point three experimental replicates were performed.

## 2-AP fluorescence

2AP fluorescent samples were excited at 315 nm (5 mm slit width), and emission was detected at 370 nm (7 mm slit width) with a FluoroMax 4 (Horiba Join Yvon Inc). Reactions contained 150 or 375 nM annealed DNA (*Table 4*) and 0.5 or 1 µM RFC in a buffer with 50 mM HEPES–NaOH pH 7.5, 200 mM NaCl, 4 mM MgCl$_2$, 1 mM TCEP and were carried out at room temperature. Experiments

**Table 4.** DNA sequences.

| Template name | Sequence | Primer name | Sequence | Name used in assay |
|---|---|---|---|---|
| Template30-20-A | TTTTTTTTTTAATGTACTCGTAGTGTCTGC | Primer20-3'abasic | GCAGACACTACGAGTACAT/3dSp/ | p/t-DNA 3'-abasic |
| | | Primer20-3'-T-phosphate | GCAGACACTACGAGTACATT/3Phos/ | p/t-DNA 3' PO4 |
| | | Primer20-3'-T-RNA | rGrCrArGrArCrArCrUrArCrGrArGrUrArCrArUrU | RNA primer/DNA template |
| | | Primer20-3'-riboT | GCAGACACTACGAGTACATrU | p/t-DNA 3' ribo |
| | | Primer20-3'-T | GCAGACACTACGAGTACATT | p/t-DNA |
| | | Primer20-2AP-0 | GCAGACACTACGAGTACAT/32AmPu/ | p/t-AP, P = 1 |
| | | Primer20-2AP-2 | GCAGACACTACGAGTAC/i2AmPr/TA | p/t-AP, P = 3 |
| Template30-T-1 | TTTTTTTTTTTTGTACTCGTAGTGTCTGC-3' | Primer20-2AP-1 | GCAGACACTACGAGTACA/i2AmPr/A | p/t-AP, P = 2 |
| Template30-20-2AP | TTTTTTTTTTT/i2AmPr/ATGTACTCGTAGTGTCTGC-3' | Primer20-3'-T | GCAGACACTACGAGTACATT | p/t-AP, t = 1 |
| Template20-5'-A | AATGTACTCGTAGTGTCTGC | Primer20-3'-T | GCAGACACTACGAGTACATT | Blunt DNA |
| | | Primer 20-3'-T-10ext | GCAGACACTACGAGTACATTTTTTTTTTTT | 3' overhang DNA |
| Template30-20-A-3'T | AATGTACTCGTAGTGTCTGCTTTTTTTTTT | Primer 20-3'-T-10ext | GCAGACACTACGAGTACATTTTTTTTTTTTT | 3' overhang dumbbell DNA |
| | TTTTTTTTTTTTTTTTTTTT | polyT 20 | TTTTTTTTTTTTTTTTTTT | ssDNA (poly T) |

(*Figure 6C*) were performed in the presence of 375 nM DNA, 0.5 µM RFC, and 2.5 µM PCNA. Experiments (*Figure 6D, E*) were performed with 150 nM DNA, 1 µM RFC, and 2.5 µM PCNA.

## Plasmid generation

The separation pin variants were introduced with site-directed mutagenesis in either pLANT-2/RIL-RFC[1 + 5] for protein purification or pRS413-RFC1 for yeast complementation. pRS413-RFC1 contains the entire RFC1 sequence, where RFC1 is expressed under the control of its own promotor.

## Yeast strains and spot assay

The genotype of the *S. cerevisiae* strain which was used in this study for transformation with the pRS413 plasmids is described in the Key Resources Table. *S. cerevisiae* culture, transformation, and tetrad dissection, were performed as previously described (*Gomes et al., 2000*).

For the spot assay, yeast grown on SC-His plate at 30°C for 2 days was inoculated into 3 ml SC-His media and grown for 3–4 hr to an OD of 0.8. Serial tenfold dilutions of the cultures starting from OD of 0.2 were plated as 4 µl drops onto YPD plates with or without chemical additives (0.01% MMS, 100 mM HU). For UV treatment, the spotted yeast was irradiated with 30 or 100 J/m$^2$ using a UVP UV Crosslinker. The plates were imaged after incubating at 18°C for 7 days, or at 30°C, 37°C for 3 days, (duplicates were done for the treatment with MMS, and triplicates for all other treatments).

## Acknowledgements

The authors thank Drs. C Xu, KK Song, and K Lee for assistance with data collection, and Drs. C Xu and A Jecrois for advice on data processing. Additionally, we thank Dr. M Hingorani for providing us with the RFC expression plasmids. We thank J Andrade and Dr. B Ueberheide for analyzing the crosslinked sample with mass spectrometry. We thank members of the Kelch, Royer, and Schiffer labs for helpful discussions. This work was funded by NIGMS (R01-GM127776). CG was supported by an Early and Advanced Postdoc Mobility (grant numbers 168972 and 177859) Fellowship of the Swiss National Science Foundation. GD was supported by LL2008 project with financial support from the MEYS CR as a part of the ERC CZ program.

## Additional information

### Funding

| Funder | Grant reference number | Author |
| --- | --- | --- |
| National Institute of General Medical Sciences | R01-GM127776-02 | Brian A Kelch |
| Schweizerischer Nationalfonds zur Förderung der Wissenschaftlichen Forschung | 177859 | Christl Gaubitz |
| Schweizerischer Nationalfonds zur Förderung der Wissenschaftlichen Forschung | 168972 | Christl Gaubitz |
| MEYS CR ERC CZ | LL2008 | Gabriel Demo |

The funders had no role in study design, data collection, and interpretation, or the decision to submit the work for publication.

### Author contributions

Christl Gaubitz, Conceptualization, Data curation, Formal analysis, Funding acquisition, Investigation, Methodology, Project administration, Validation, Visualization, Writing – original draft, Writing – review and editing; Xingchen Liu, Formal analysis, Investigation, Writing – original draft, Writing – review and

editing; Joshua Pajak, Formal analysis, Writing – original draft, Writing – review and editing; Nicholas P Stone, Janelle A Hayes, Gabriel Demo, Formal analysis; Brian A Kelch, Conceptualization, Data curation, Formal analysis, Funding acquisition, Methodology, Project administration, Supervision, Validation, Visualization, Writing – original draft, Writing – review and editing

**Author ORCIDs**
Christl Gaubitz  http://orcid.org/0000-0002-6047-9282
Xingchen Liu  http://orcid.org/0000-0002-9089-1761
Joshua Pajak  http://orcid.org/0000-0001-5781-0870
Nicholas P Stone  http://orcid.org/0000-0002-5869-0329
Brian A Kelch  http://orcid.org/0000-0002-1369-6989

**Decision letter and Author response**
Decision letter https://doi.org/10.7554/eLife.74175.sa1
Author response https://doi.org/10.7554/eLife.74175.sa2

---

## Additional files

### Supplementary files
• Transparent reporting form

### Data availability
All coordinates and cryoEM maps were deposited in the PDB and EMDB during revision.

The following datasets were generated:

| Author(s) | Year | Dataset title | Dataset URL | Database and Identifier |
|---|---|---|---|---|
| Gaubitz C, Liu X, Pajak J, Stone NP, Hayes JA, Demo G, Kelch BA | 2021 | Structure of the yeast clamp loader (Replication Factor C RFC) bound to the sliding clamp (Proliferating Cell Nuclear Antigen PCNA) in an autoinhibited conformation | https://www.ebi.ac.uk/emdb/EMD-25568 | EMDB, EMD-25568 |
| Gaubitz C, Liu X, Pajak J, Stone NP, Hayes JA, Demo G, Kelch BA | 2022 | Structure of the yeast clamp loader (Replication Factor C RFC) bound to the sliding clamp (Proliferating Cell Nuclear Antigen PCNA) in an autoinhibited conformation | https://doi.org/10.2210/pdb7THJ/pdb | Worldwide Protein Data Bank, 10.2210/pdb7THJ/pdb |
| Gaubitz C, Liu X, Pajak J, Stone NP, Hayes JA, Demo G, Kelch BA | 2022 | Structure of the yeast clamp loader (Replication Factor C RFC) bound to the sliding clamp (Proliferating Cell Nuclear Antigen PCNA) in an autoinhibited conformation | https://www.ebi.ac.uk/emdb/EMD-25569 | EMDB, EMD-25569 |
| Gaubitz C, Liu X, Pajak J, Stone NP, Hayes JA, Demo G, Kelch BA | 2022 | Structure of the yeast clamp loader (Replication Factor C RFC) bound to the sliding clamp (Proliferating Cell Nuclear Antigen PCNA) in an autoinhibited conformation | https://doi.org/10.2210/pdb7TIC/pdb | Worldwide Protein Data Bank, 10.2210/pdb7TIC/pdb |

*Continued on next page*

*Continued*

| Author(s) | Year | Dataset title | Dataset URL | Database and Identifier |
|---|---|---|---|---|
| Gaubitz C, Liu X, Pajak J, Stone NP, Hayes JA, Demo G, Kelch BA | 2022 | Structure of the yeast clamp loader (Replication Factor C RFC) bound to the sliding clamp (Proliferating Cell Nuclear Antigen PCNA) in an autoinhibited conformation | https://www.ebi.ac.uk/emdb/EMD-25614 | EMDB, EMD-25614 |
| Gaubitz C, Liu X, Pajak J, Stone NP, Hayes JA, Demo G, Kelch BA | 2022 | Structure of the yeast clamp loader (Replication Factor C RFC) bound to the sliding clamp (Proliferating Cell Nuclear Antigen PCNA) in an autoinhibited conformation | https://doi.org/10.2210/pdb7THV/pdb | Worldwide Protein Data Bank, 10.2210/pdb7THV/pdb |
| Gaubitz C, Liu X, Pajak J, Stone NP, Hayes JA, Demo G, Kelch BA | 2022 | Structure of the yeast clamp loader (Replication Factor C RFC) bound to the open sliding clamp (Proliferating Cell Nuclear Antigen PCNA) | https://www.ebi.ac.uk/emdb/EMD-25615 | EMDB, 25615 |
| Gaubitz C, Liu X, Pajak J, Stone NP, Hayes JA, Demo G, Kelch BA | 2022 | Structure of the yeast clamp loader (Replication Factor C RFC) bound to the open sliding clamp (Proliferating Cell Nuclear Antigen PCNA) | https://doi.org/10.2210/pdb7TKU/pdb | Worldwide Protein Data Bank, 10.2210/pdb7TKU/pdb |
| Gaubitz C, Liu X, Pajak J, Stone NP, Hayes JA, Demo G, Kelch BA | 2022 | Structure of the yeast clamp loader (Replication Factor C RFC) bound to the open sliding clamp (Proliferating Cell Nuclear Antigen PCNA) | https://www.ebi.ac.uk/emdb/EMD-25753 | EMDB, EMD-25753 |
| Gaubitz C, Liu X, Pajak J, Stone NP, Hayes JA, Demo G, Kelch BA | 2022 | Structure of the yeast clamp loader (Replication Factor C RFC) bound to the open sliding clamp (Proliferating Cell Nuclear Antigen PCNA) | https://doi.org/10.2210/pdb7TI8/pdb | Worldwide Protein Data Bank, 10.2210/pdb7TI8/pdb |
| Gaubitz C, Liu X, Pajak J, Stone NP, Hayes JA, Demo G, Kelch BA | 2022 | Structure of the yeast clamp loader (Replication Factor C RFC) bound to the open sliding clamp (Proliferating Cell Nuclear Antigen PCNA) and primer-template DNA | https://www.ebi.ac.uk/emdb/EMD-25616 | EMDB, EMD-25616 |
| Gaubitz C, Liu X, Pajak J, Stone NP, Hayes JA, Demo G, Kelch BA | 2022 | Structure of the yeast clamp loader (Replication Factor C RFC) bound to the open sliding clamp (Proliferating Cell Nuclear Antigen PCNA) and primer-template DNA | http://dx.doi.org/10.2210/pdb7tib/pdb | Worldwide Protein Data Bank, 10.2210/pdb7tib/pdb |
| Gaubitz C, Liu X, Pajak J, Stone NP, Hayes JA, Demo G, Kelch BA | 2022 | Structure of the yeast clamp loader (Replication Factor C RFC) bound to the sliding clamp (Proliferating Cell Nuclear Antigen PCNA) and primer-template DNA | https://www.ebi.ac.uk/emdb/EMD-25617 | EMDB, EMD-25617 |

*Continued on next page*

*Continued*

| Author(s) | Year | Dataset title | Dataset URL | Database and Identifier |
|---|---|---|---|---|
| Gaubitz C, Liu X, Pajak J, Stone NP, Hayes JA, Demo G, Kelch BA | 2022 | Structure of the yeast clamp loader (Replication Factor C RFC) bound to the sliding clamp (Proliferating Cell Nuclear Antigen PCNA) and primer-template DNA | https://doi.org/10.2210/pdb7TID/pdb | Worldwide Protein Data Bank, 10.2210/pdb7TID/pdb |

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
