## [Editor Report]

Clamp loader-sliding clamp complexes are required for DNA replication and repair in all domains of life. This study reports several cryo-EM structures of multiple clamp loading intermediates from a single species, *Saccharomyces cerevisiae*, providing new insights into the mechanism of clamp loading and the ligand-induced conformational dynamics of molecular motors and switches.

---

## [Decision Letter]

**Decision letter after peer review:**

Thank you for submitting your article "Cryo-EM structures reveal high-resolution mechanism of a DNA polymerase sliding clamp loader" for consideration by *eLife*. Your article has been reviewed by 2 peer reviewers, one of whom is a member of our Board of Reviewing Editors and the evaluation has been overseen by Volker Dötsch as the Senior Editor. The following individual involved in review of your submission has agreed to reveal their identity: Linda Bloom (Reviewer #2).

The reviewers have discussed their reviews with one another, and the Reviewing Editor has drafted this integrated review to help you prepare a revised submission.

Essential revisions:

1) The authors conclude that clamp closing does not require ATP hydrolysis. However, kinetic data in the literature support a mechanism in which the clamp loader hydrolyzes ATP prior to clamp closing. In the absence of supporting kinetic data, it may be overinterpreting structural data (of crosslinked samples) to assert that the clamp loader does not need to hydrolyze ATP prior to closing the clamp. We recommend the authors tone down their interpretation or provide kinetic experiments that substantiate this claim.

2) The authors identify a separation pin that melts the final base pair of the primer/template duplex and flips the 3' base of the primer strand. The authors speculate this mechanism could be used for distinguishing p/t substrates from other DNA structures, but the importance of the separation pin for clamp loader function and the physiological significance of p/t melting and base flipping by RFC remain unclear. To substantiate the claim, we ask that the authors include DNA binding data of WT and separation pin mutant complexes (F582A and W638G), examining their affinity and DNA substrate specificity.

With respect to the effect of separation pin mutations on ATPase activity, we think that a statistical analysis of the differences in the ATPase activities of wild-type and mutant clamp loaders would be helpful for determining whether separation pin mutations have an effect on the activity. Moreover, steady-state ATPase activity was measured in this experiment and these rates may not reveal differences in rates of intermediate steps in the clamp loading reaction. For the mutations to affect the ATPase activity, they would have to either change the rate of the rate-limiting step in the pathway or change the identity of the rate-limiting step. Thus, the decrease in ATPase activity for W638G mutant could be interesting if statistically significant.

Perhaps characterization of a double F582A/W638G mutant would also shed more light on the importance of this region.

3) There appear to be discrepancies between the reported resolutions of the RFC:PCNA-DNA-open and RFC:PCNA-DNA-closed structures in Table S2 and those seen in the FSC curves in Figure S1D. Resolutions of 3.4 and 3.3A should yield 0.143 FSC at 0.29 and 0.3 1/A, but the values in the FSC curves are lower than that. Uniform scaling of the x-axes should also be used. Related to this point, the FSC curves indicate that the RFC:PCNA-DNA-open and RFC:PCNA-DNA-closed models may have been overfitted to the cryo-EM map. The 1/A values at 0.5 FSC for model vs. map should be the same or slightly lower than the 1/A values at 0.143 FSC for the masked maps and not higher.

---

## [Author Response]

Essential revisions:1) The authors conclude that clamp closing does not require ATP hydrolysis. However, kinetic data in the literature support a mechanism in which the clamp loader hydrolyzes ATP prior to clamp closing. In the absence of supporting kinetic data, it may be overinterpreting structural data (of crosslinked samples) to assert that the clamp loader does not need to hydrolyze ATP prior to closing the clamp. We recommend the authors tone down their interpretation or provide kinetic experiments that substantiate this claim.

We agree with the reviewers that the kinetic data (particularly from the Bloom lab) show that, under normal conditions, ATP hydrolysis precedes clamp closure. We have adjusted our discussion and our model in Figure 7 to place our findings in context of that previous data. In short, we speculate that ATPγS slows down hydrolysis sufficiently that clamp closure occurs prior to hydrolysis. In this context, our data show that ATP hydrolysis is not strictly required for clamp closure, but that under normal conditions ATP hydrolysis precedes clamp closure. We are currently designing experiments to test this hypothesis.

2) The authors identify a separation pin that melts the final base pair of the primer/template duplex and flips the 3' base of the primer strand. The authors speculate this mechanism could be used for distinguishing p/t substrates from other DNA structures, but the importance of the separation pin for clamp loader function and the physiological significance of p/t melting and base flipping by RFC remain unclear. To substantiate the claim, we ask that the authors include DNA binding data of WT and separation pin mutant complexes (F582A and W638G), examining their affinity and DNA substrate specificity.With respect to the effect of separation pin mutations on ATPase activity, we think that a statistical analysis of the differences in the ATPase activities of wild-type and mutant clamp loaders would be helpful for determining whether separation pin mutations have an effect on the activity. Moreover, steady-state ATPase activity was measured in this experiment and these rates may not reveal differences in rates of intermediate steps in the clamp loading reaction. For the mutations to affect the ATPase activity, they would have to either change the rate of the rate-limiting step in the pathway or change the identity of the rate-limiting step. Thus, the decrease in ATPase activity for W638G mutant could be interesting if statistically significant.Perhaps characterization of a double F582A/W638G mutant would also shed more light on the importance of this region.

We agree with the reviewers that this surprising aspect of RFC activity is of particular interest. We now report a more rigorous statistical treatment of the ATPase data. These results confirm that the W638G variant has significantly less steady-state ATPase activity than WT, although the reduction is not particularly dramatic (Figure 6, Figure 6 —figure supplement 1). The rate-limiting step, as measured in rapid kinetics experiments by the Bloom and Hingorani groups, has been found to be clamp release, so perhaps it is unsurprising that these mutations do not dramatically affect steady-state ATPase activity. To probe this question in more detail, it would require pre-steady state kinetics, which we feel is beyond the scope of this manuscript. We would be happy to collaborate with world experts to perform these studies in the future. We discuss these findings in greater detail in the text.

We have measured apparent Kd’s by titrating DNA and measuring ATPase activity (Figure 6 —figure supplement1). These experiments show that the F582A variant has similar affinity for DNA and that the W638G has a slightly reduced affinity for DNA relative to WT.

Furthermore, we now report experiments measuring ATPase activity in the presence of various DNA substrates (Figure 6 —figure supplement 2). However, none show any substantial difference relative to WT (i.e. the activity profiles are nearly identical), suggesting that our hypothesis that the separation pin acts to discriminate against certain moieties at the primer-template junction is either wrong or incomplete. We have updated the text to reflect these new findings.

Because the reduction in ATPase activity is not particularly dramatic, even with a dramatic reduction in base-flipping, we sought to probe the physiological function of this base-flipping mechanism in more detail. We now report functional characterization of these mutations in *S. cerevisae*. To our surprise, FA and WG mutations have no effect on yeast growth when present as the only copy of RFC1. This is true under a variety of growth conditions, such as altered temperature, and treatment with damaging agents such as MMS, HU, and UV radiation (Figure 6 —figure supplement 3). Although we have found no clear phenotype for this base-flipping, we speculate that it carries an as-yet-unknown physiological importance because it has been conserved across all eukarya. Combined with our biochemical data, the role of the separation pin and its base-flipping activity remains mysterious. Because it is likely to require substantially more work to discern its role (both in terms of biochemical and genetic studies), we will address this in future studies. We have updated the text and added figures (Figure 6 —figure supplements 2 and 3) to reflect our new results.

3) There appear to be discrepancies between the reported resolutions of the RFC:PCNA-DNA-open and RFC:PCNA-DNA-closed structures in Table S2 and those seen in the FSC curves in Figure S1D. Resolutions of 3.4 and 3.3A should yield 0.143 FSC at 0.29 and 0.3 1/A, but the values in the FSC curves are lower than that. Uniform scaling of the x-axes should also be used. Related to this point, the FSC curves indicate that the RFC:PCNA-DNA-open and RFC:PCNA-DNA-closed models may have been overfitted to the cryo-EM map. The 1/A values at 0.5 FSC for model vs. map should be the same or slightly lower than the 1/A values at 0.143 FSC for the masked maps and not higher.

We thank the reviewer for pointing out this oversight. The reviewer is right, the FSC curves of RFC with DNA^PCNA-open^ and RFC DNA^PCNA-closed^ structures were not correct, and we now corrected this error. We also scaled the axes uniformly. Moreover, we repeated the last round of refinement using weight optimization for all structures to avoid overfitting, which led to improvement of model geometry and lowered the 1/ Å values at 0.5 FSC for model vs. map (Table 3 and Figure 1 —figure supplement 1). We have submitted all maps and models to the EMDB and PDB, and we included the accession codes as well as the updated model statistics in Table 3.